# Quasi-Self-Concordant Optimization with Lewis Weights

**Alina Ene**
Department of Computer Science
Boston University
aene@bu.edu

**Ta Duy Nguyen**
Department of Computer Science
Boston University
taduy@bu.edu

**Adrian Vladu**
CNRS & IRIF
Université Paris Cité
vladu@irif.fr

## Abstract

In this paper, we study the problem $\min_{x \in \mathbb{R}^d, Nx=v} \sum_{i=1}^n f((Ax-b)_i)$ for a quasi-self-concordant function $f : \mathbb{R} \to \mathbb{R}$, where $A, N$ are $n \times d$ and $m \times d$ matrices, $b, v$ are vectors of length $n$ and $m$ with $n \geq d$. We show an algorithm based on a trust-region method with an oracle that can be implemented using $\widetilde{O}(d^{1/3})$ linear system solves, improving the $\widetilde{O}(n^{1/3})$ oracle by [Adil-Bullins-Sachdeva, NeurIPS 2021]. Our implementation of the oracle relies on solving the overdetermined $\ell_\infty$-regression problem $\min_{x \in \mathbb{R}^d, Nx=v} \|Ax-b\|_\infty$. We provide an algorithm that finds a $(1+\varepsilon)$-approximate solution to this problem using $O((d^{1/3}/\varepsilon + 1/\varepsilon^2)\log(n/\varepsilon))$ linear system solves. This algorithm leverages $\ell_\infty$ Lewis weight overestimates and achieves this iteration complexity via a simple lightweight IRLS approach, inspired by the work of [Ene-Vladu, ICML 2019]. Experimentally, we demonstrate that our algorithm significantly improves the runtime of the standard CVX solver.

## 1 Introduction

Quasi-self-concordant (QSC) optimization encompasses a broad class of problems that have long been central to convex optimization, numerical analysis, robust statistics, data fitting, and constrained optimization [Bac10, KSJ18, MFBR19, STD19, CJJ+20, CHJ+22, Doi23]. Notable special cases include logistic regression, softmax regression, and regularized $\ell_p$ regression, which have been widely studied due to their broad range of applications in machine learning.

Due to their favorable properties, QSC functions have enjoyed efficient optimization methods which require strictly weaker curvature assumptions than, for example, strong convexity. Yet, without strong promises about the underlying model, QSC optimization is still a challenging problem for which first order methods only provide low precision solution with $\text{poly}(1/\varepsilon)$ convergence rate as opposed to the desirable $\text{poly}\log(1/\varepsilon)$ rate, where $\varepsilon$ is the sub-optimality gap, while typical second order methods are computationally expensive. Thus significant challenges remain for the pursuit of more efficient optimization algorithms for this class of functions.

In this work, we study the constrained QSC optimization problem

$$\min_{x \in \mathbb{R}^d, Nx=v} h(x) := \sum_{i=1}^n f\left((Ax-b)_i\right), \tag{1}$$

where $f : \mathbb{R} \to \mathbb{R}$ is a QSC function, $A \in \mathbb{R}^{n \times d}$, $N \in \mathbb{R}^{m \times d}$ are matrices, and $b \in \mathbb{R}^n$, $v \in \mathbb{R}^m$ are vectors. Each term $f\left((Ax-b)_i\right)$ can be viewed as measuring the fit to a single datapoint. We focus in particular on the overdetermined regime where the number of datapoints $n$ is significantly larger than the number of variables $d$. This setting is central to large-scale data analysis. In regression tasks, standard techniques for managing large $n$ include data sparsification [SS08, CLM+15], subspace

39th Conference on Neural Information Processing Systems (NeurIPS 2025).

embeddings [NN13], and tools from randomized numerical linear algebra [DM18]. However, for QSC objectives, existing approaches typically exhibit iteration complexity that scales with $n$ rather than the intrinsic dimension $d$, which makes them poorly suited to modern high-sample regimes. Thus, the central question to our work is:

*Can we design an algorithm for Problem (1) with iteration complexity nearly independent of $n$?*

We address this question by showing that QSC functions can be minimized to high precision ($\mathrm{poly}\log(1/\varepsilon)$ convergence rate) in $\tilde{O}(d^{1/3})$ iterations, each of which makes a constant number of calls to a linear system solver involving structured $d \times d$ matrices, thereby drastically improving efficiency and scalability of existing approaches. Our approach is based on a trust-region method, for which we show a fast oracle implementation. Our oracle implementation relies on solving $\ell_\infty$ regression problems. Instead of leveraging data sparsification to reduce problem size [CP15, JLLS23, JLLS24, WY24], which often entails high computational costs when $n$ large, our algorithm is inspired by the recent advances on general $\ell_p$ regression, such as the work of Jambulapati-Liu-Sidford [JLS22]. We introduce a new approach that leverages $\ell_\infty$ Lewis weights inside a simple, lightweight Iteratively Reweighted Least Squares (IRLS) algorithm [EV19], which is of independent interest. A simple modification of our algorithm can be used to solve QSC optimization problems in the underdetermined regime $n \leq d$ as well, and it provides a much simpler and practical algorithm with theoretical guarantees that mirror those of the state of the art algorithm of [ABS21] that currently lacks a practical implementation.

## 1.1 Our contributions

We consider the QSC optimization problem (1), where $f$ is a general $C$-QSC function (Definition 2.1), potentially non-smooth and not strongly convex. Following the literature on regression algorithms [APS19, AKPS19, JLS22] and the prior work by [ABS21], we focus on designing highly efficient iterative algorithms that rely on (structured) linear system solvers. Throughout this paper, we define one iteration as a single call to the linear system solver. Since the linear system solves dominate the running time in all algorithms we discuss, the total runtime is essentially the product of the iteration count and the cost of solving these systems.

Our first main contribution is a new trust-region method for affine-constrained $C$-QSC optimization that achieves the following performance guarantee.

**Theorem 1.1.** *Let $f : \mathbb{R} \to \mathbb{R}$ be a $C$-quasi-self-concordant function, let $A \in \mathbb{R}^{n \times d}$, $N \in \mathbb{R}^{m \times d}$, $b \in \mathbb{R}^n$ and $v \in \mathbb{R}^m$ with $n > d$. Define the function $h : \mathbb{R}^d \to \mathbb{R}$ as $h(x) = \sum_{i=1}^n f\left((Ax - b)_i\right)$. Let $x^{(0)}$ be an initial solution satisfying $Nx^{(0)} = v$, and let $R$ be any value such that $\max_{x \in \left\{x : h(x) \leq h\left(x^{(0)}\right)\right\}} \|Ax - Ax^*\|_\infty \leq R$. Suppose $h$ is bounded from below, and let $B$ be any value satisfying $h(x) \geq B$ for all $x \in \mathbb{R}^d$. Then there exists an algorithm that, given $x^{(0)}$, $R$, and $B$ as input, it computes an $\varepsilon$-additive approximation to the problem $\min_{x \in \mathbb{R}^d, Nx=v} h(x)$ by solving $O\left(CR \log\left(CR\right) \log^2\left(\frac{h(x^{(0)}) - h(x^*)}{\varepsilon}\right)\right)$ subproblems, each of which makes $O\left(d^{1/3} \log n\right)$ calls to a linear system solver with matrices of the form $A^\top D A$ and $N\left(A^\top DA\right)^+ N^\top$, where $D$ is a positive diagonal matrix.*

**Comparison to prior works**: Prior to our work, the algorithm with state of the art iteration complexity for general (non-smooth) QSC optimization was given by [ABS21], which achieved iteration complexity $O\left(n^{1/3} \log^{O(1)}(n) \cdot CR \log(CR) \log\left(\frac{h(x^{(0)}) - h(x^*)}{\varepsilon}\right)\right)$. Our algorithm improves this to a dependence on $d^{1/3}$, which is a substantial improvement in the overdetermined regime $n \gg d$. Moreover, the algorithm of [ABS21] is complex and makes use of several parameters set to guarantee the theoretical runtime. In practice, to obtain an efficient algorithm, these parameters need to be carefully tuned and currently this algorithm of [ABS21] lacks a practical implementation for this reason. On the other hand, our algorithm is significantly simpler and implementable in practice, with an excellent empirical performance compared to CVX.

Our algorithm builds on a trust-region based algorithmic framework developed in prior work [AZLOW17, CMTV17, CJJ$^+$20, ABS21]. An important property of QSC functions shown by [KSJ18] is that the Hessian at a point $x$ is relatively stable in a region around $x$. Prior works such as [CJJ$^+$20, ABS21] build on this property and design algorithms that iteratively minimize a local

second-order approximation of the function in a region where the Hessian is sufficiently stable. The key difficulty in implementing this approach lies in designing efficient algorithms for solving the resulting subproblems. One of our main contributions is a new efficient subroutine for solving the resulting subproblems. We develop our subroutine by first designing a novel and simple algorithm for solving $\ell_\infty$ regression, which is of independent interest. Our subroutine algorithm then solves $\ell_2$-regularized $\ell_\infty$ regression problems, enjoying the simplicity and efficiency of our $\ell_\infty$ regression solver. The following theorem states the guarantees for our $\ell_\infty$ regression algorithm.

**Theorem 1.2.** *Let $A \in \mathbb{R}^{n \times d}$, $N \in \mathbb{R}^{m \times d}$, $b \in \mathbb{R}^n$ and $v \in \mathbb{R}^m$, with $n > d$, and let $\varepsilon > 0$ be a scalar. There exists an algorithm which provides a $(1 + \varepsilon)$-multiplicative approximation to the $\ell_\infty$ regression problem $\min_{x \in \mathbb{R}^d, Nx=v} \|Ax - b\|_\infty$ by solving $O\left(\log \frac{\log(n)}{\varepsilon}\right)$ subproblems, each of which makes $O\left(\left(\frac{d^{1/3}}{\varepsilon} + \frac{1}{\varepsilon^2}\right) \log \frac{n}{\varepsilon}\right)$ calls to a solver for structured linear systems involving matrices of the form $A^\top D A$ and $N\left(A^\top D A\right)^+ N^\top$, where $D$ is a positive diagonal matrix.*

Our algorithm for $\ell_\infty$ regression achieves state of the art iteration complexity (in terms of dependence on the dimension $d$) via an iteratively reweighted least squares (IRLS) approach that solves a weighted least squares instance in each iteration, whose solution is provided by a single linear system solve. IRLS algorithms are favored in practice due to their simplicity and ease of implementation. IRLS algorithms were first introduced in pioneering work from the 1960s [Law61, RU68]. Subsequently, there has been significant interest in developing IRLS methods with provable convergence guarantees [SV16b, SV16a, EV19, APS19].

**Comparison to prior works**: Prior to our work, the state of the art IRLS algorithm for $\ell_\infty$ regression is the algorithm of [EV19]. The work [JLS22] gives a non-IRLS algorithm for $\ell_\infty$ regression with the state of the art iteration complexity $\widetilde{O}\left(d^{1/3}/\varepsilon^{2/3}\right)$. We highlight the differences between our algorithm and these works. The algorithm of [JLS22] works by leveraging $\ell_\infty$ Lewis weights, and uses a reduction to the Monteiro-Svaiter accelerated gradient descent algorithm by [CJJ+20]. Monteiro-Svaiter acceleration is a complex scheme, requiring to solve an implicit equation in each iteration. While an implementation is later provided in [CHJ+22], the practicality of Monteiro-Svaiter acceleration remains an open question [CHJ+22]. Our $\ell_\infty$ regression algorithm also uses Lewis weights, but from a completely different approach, based on an IRLS algorithm. The starting point of our algorithm is the algorithmic framework by [EV19]. However the algorithm by [EV19] is very specific to the uniform initialization of the solution and only achieves iteration complexity $\widetilde{O}\left(n^{1/3}/\varepsilon^{2/3}\right)$. The non-uniform initialization via $\ell_\infty$ Lewis weights in our algorithm immediately breaks the analysis by [EV19]. To overcome this barrier requires new insights. Our algorithm also further improves the iteration complexity of [EV19]'s algorithm to $\widetilde{O}\left(d^{1/3}/\varepsilon + 1/\varepsilon^2\right)$ while retaining the simplicity and practicality of an IRLS method.

**Underdetermined regime** $n \leq d$: A simple modification of our algorithm can be used in the regime $n \leq d$ and it obtains a guarantee which is analogous to Theorem 1.1 but with an iteration complexity that depends on the smaller dimension $n$. Our algorithm matches the iteration complexity of the state of the art algorithm of [ABS21] up to poly-logarithmic factors, while being significantly simpler and more practical. As discussed above, our algorithm is an IRLS based method where the main computation in each iteration is a weighted least squares regression, which is beneficial in practice.

**Further extensions:** It is worth noting that while we followed the formulation from [ABS21], our solver can handle more general functions provided that their Hessian is stable within any $\ell_\infty$ box of fixed radius. This allows, for example, to minimize functions of the form $\sum f_i\left((Ax - b)_i\right)$ in both the constrained and unconstrained settings, where each $f_i$ in the decomposition has a bounded third derivative. The same framework applies to functions without the decomposable structure, provided that a similar stability property holds.

## 1.2 Related work

The class of quasi-self-concordant (QSC) functions was introduced in [Bac10], extending the definition of standard self-concordant functions to logistic regression models. As QSC functions encompass various other important models, such as softmax and $\ell_p$ regressions, they have been further explored in a series of influential works [STD19, KSJ18, CJJ+20, ABS21, Doi23].

Most of these studies rely on trust-region Newton methods, which inherently introduce a dependence on the number of iterations dictated by the shape and size of the trust region. Typically, these regions are $\ell_2$ balls centered at the current iterate, leading to convergence guarantees expressed in terms of the $\ell_2$ diameter of the search region. Notably, [ABS21] applied the techniques of [AZLOW17, CMTV17] to instead establish guarantees in terms of the $\ell_\infty$ diameter of the search region by solving a sequence of second-order subproblems with $\ell_\infty$ constraints. Our algorithm for QSC optimization follows a similar principle.

A significant contribution by [Doi23] introduced acceleration techniques to achieve an iteration complexity of $\widetilde{O}((CR)^{2/3})$, where $R$ denotes the $\ell_2$ diameter of the domain. However, since the worst-case ratio between the $\ell_2$ and $\ell_\infty$ diameters is $\sqrt{n}$, the iteration complexity may degrade by a factor of $n^{1/3}$ when measured under the $\ell_\infty$ benchmark used in [ABS21] and in this work—while still maintaining the desirable sublinear dependence on $C$ and $R$. In comparison, our approach incurs only a $d^{1/3}$ factor, which represents a substantial improvement when $n \gg d$. Additionally we note that accelerated algorithms typically require smoothness, and this is the case for those given in [CJJ+20, Doi23]. However, both the algorithm from [ABS21] and ours manage to achieve an improved iteration complexity, without resorting to smoothness or other additional properties. We note that [KSJ18] provide an analysis for both the smooth and non-smooth settings, but they exhibit a dependence on the $\ell_2$ diameter of the level set of the initialization point, which scales as $n^{1/2}$ in the worst case.

$\ell_\infty$ regression is a classical optimization problem, but the first major breakthrough occurred in the special case of max flow in [CKM+11], where the authors developed an algorithm requiring $\widetilde{O}\left(n^{1/3}\right)$ linear system solves, improving upon the standard $\widetilde{O}\left(n^{1/2}\right)$ complexity. This result was further refined and generalized in [CMMP13, EV19], both achieving $\widetilde{O}\left(n^{1/3}\right)$ iteration complexity. More recently, [JLS22] studied both $\ell_p$ regression for general $p < \infty$ and $\ell_\infty$ regression, presenting algorithms whose iteration count depends on $d$ rather than $n$, leveraging Lewis weights. Previously, [LS14] devised an interior point method for tall LPs where the number of iterations depends on $d$ rather than $n$ by employing Lewis weights. For the specific case of $\ell_\infty$ regression, [JLS22] incorporated Lewis weights into the accelerated second-order method of [CJJ+20], which approximates the $\ell_\infty$ norm via the Hessian of a log-exp-sum function. In contrast, while we also draw on the Lewis weight approach from [JLS22], we develop a simpler, lightweight IRLS method inspired by [EV19].

### 1.3 Overview of our approach

To establish our main result, we employ a trust-region method. While such methods have been extensively studied, the distinguishing feature of our approach is that the convergence rate depends on the $\ell_\infty$ diameter of the level set that contains the initialization point, rather than its $\ell_2$ diameter, which can be larger by up to a factor of $\sqrt{n}$. This yields both theoretical and practical improvements. Our method is motivated by the box-constrained Newton method which was introduced by [CMTV17, AZLOW17], and later adapted to quasi-self-concordant minimization [ABS21]. This perspective allows us to reduce the optimization process to executing an outer loop of $O\left(CR \log \frac{h(x_0) - h(x^*)}{\varepsilon}\right)$ iterations, where $C$ is the QSC parameter of $f$ and $R = \max\{\|Ax - Ax^*\|_\infty : h(x) \leq h(x_0)\}$ denotes the $\ell_\infty$ diameter of the level set of the initialization point $x_0$.

In each iteration of the outer loop, we compute the update by approximately minimizing a quadratic approximation of $h$ over the $\ell_\infty$ ball of radius $1/C$ centered at the current iterate. This radius constraint guarantees that the Hessian of $h$ remains well approximated within the entire trust region. Consequently, the approximate minimizer computed here achieves sufficient decrease in objective value compared to the true minimizer of the region.

We then show that the required subproblem is of the form

$$\max_{\|A\Delta\|_\infty \leq \frac{1}{C}} g^\top (A\Delta) - \frac{1}{2} (A\Delta)^\top D (A\Delta) , \tag{2}$$

where $g \in \mathbb{R}^m$ is a vector and $D$ is a non-negative diagonal matrix and can be reduced to implementing a residual solver which approximately maximizes a concave quadratic objective as in (2) subject to $\ell_\infty$ constraints. We describe this reduction in Section 4. To obtain an efficient residual solver, we design an IRLS method whose convergence rate depends only on the smaller dimension $d$, requiring

only $\widetilde{O}\left(d^{1/3}\right)$ linear system solves. Finally, the extra factor $\log\left(CR\right)\log\left(\frac{h(x^{(0)})-h(x^*)}{\varepsilon}\right)$ comes from a binary search procedure, detailed in Section 4.

## 2    Preliminaries

**Notation**. Throughout the paper, when it is clear from context, we use scalar operations between vectors to denote the coordinate-wise application, e.g, for $x \in \mathbb{R}^d$, $x^2$ is an $\mathbb{R}^d$ vector with entry $x_i^2$ in the $i$-th coordinate. For $w \in \mathbb{R}^d$, we use $\mathrm{diag}\left(w\right) \in \mathbb{R}^{d \times d}$ to denote the diagonal matrix with diagonal entries given by $w$. When it is clear from context, we also abuse the notation and use the scalar $a$ to denote the vector with all coordinates having value $a$, and the dimension will be inferred from the context.

**Quasi-self-concordant functions**. We recall the definition of quasi-self-concordant functions.

**Definition 2.1.** A function $f : \mathbb{R} \to \mathbb{R}$, is $C$-quasi-self-concordant for $C > 0$ if for all $x \in \mathbb{R}$, $|f'''(x)| \leq C f''(x)$.

**Lewis weights and computation**. Our main algorithms rely on the use of $\ell_\infty$ Lewis weight overestimates. To start, we introduce leverage scores and $\ell_\infty$ Lewis weights for a matrix $A \in \mathbb{R}^{n \times d}$.

**Definition 2.2** (Leverage scores). For a matrix $A$, the leverage score of the $i$-th row of $A$ is given by $\sigma\left(A\right)_i = a_i^\top \left(A^\top A\right)^{-1} a_i$ where $a_i$ is the $i$-th row of $A$.

**Definition 2.3** ($\ell_\infty$ Lewis weights). The $\ell_\infty$ Lewis weights of matrix $A$ are the unique vector $w \in \mathbb{R}_{\geq 0}^n$ that satisfies $w_i = \sigma\left(\mathrm{diag}(w)^{1/2}A\right)_i$ for all $i \in [n]$.

While finding $\ell_\infty$ Lewis weights is computationally expensive, for the purpose of our algorithms, we only need to use $\ell_\infty$ Lewis weight overestimates, a notion introduced by [JLS22].

**Definition 2.4** ($\ell_\infty$ Lewis weight overestimates). The $\ell_\infty$ Lewis weight overestimates of matrix $A$ are a vector $w \in \mathbb{R}_{\geq 0}^n$ that satisfies $d \leq \|w\|_1 \leq 2d$ and $w_i \geq \sigma\left(\mathrm{diag}(w)^{1/2}A\right)_i$ for all $i \in [n]$.

One can efficiently compute $\ell_\infty$ Lewis weight overestimates of matrix $A$ by the fixed point iterations by [CCLY19], using $\widetilde{O}(1)$ linear system solves (see also [JLS22]).

**Linear system solver**. Our algorithms in the unconstrained setting assume access to an oracle that solves problems of the form $\min_{x:g^\top x=-1}\langle r, (Ax)^2\rangle$. Finding the minimizer $x$ is equivalent to solving a linear system of the form $A^\top DAx = g$, for a diagonal matrix $D$. The problem has a closed form solution $x = -\frac{B^{-1}g}{g^\top B^{-1}g}$ where $B = A^\top \mathrm{diag}(r)A$.

**Energy increase lower bounds**. Our approach is based on the electrical flow interpretation of the problem. Electrical flow interpretation was developed for flow problems on graph, and later extended for problems involving general linear systems. For vector $g \in \mathbb{R}^d$, matrix $A \in \mathbb{R}^{n \times d}$, and vector $r \in \mathbb{R}_{\geq 0}^n$ which we will refer to as resistances, we consider the following quantity $\mathcal{E}(r) = \min_{x:g^\top x=-1}\langle r, (Ax)^2\rangle$, which we will refer to as the electrical energy. $\mathcal{E}(r)$ has the nice property of being monotone in $r$. The increase in the energy when the resistances increase is lower bounded in the two lemmas A.1 and A.2 (provided in the appendix), which are both critical in the design and analysis of our algorithms.

## 3    $\ell_\infty$-Regression with Lewis Weights

As a warm up for the general QSC optimization algorithm, in this section, we solve the problem of overdetermined $\ell_\infty$-regression to a $(1 + \varepsilon)$-approximation, in the form

$$\min_{g^\top x=-1}\|Ax\|_\infty \tag{3}$$

where $A \in \mathbb{R}^{n \times d}$ with $n \geq d$. That is, we want to find a solution $x$ to the above problem that satisfies $g^\top x = -1$ and $\|Ax\|_\infty \leq (1 + \varepsilon)\|Ax^*\|_\infty$, where we denote $x^* = \arg\min_{g^\top x=-1}\|Ax\|_\infty$. We note that the commonly seen problem $\min_x \|Ax - b\|_\infty$ can be transformed into the above form by adding $b$ as a column to $A$ and considering $x \in \mathbb{R}^{d+1}$, with the extra dimension having value $-1$.

---

**Algorithm 1** $\ell_\infty$-regression for $\min_{g^\top x=-1} \|Ax\|_\infty$

---

1: **Input**: $A$, $g$, $\varepsilon$
2: **Output**: Find $x$ such that $g^\top x = -1$ and $\|Ax\|_\infty \leq (1+\varepsilon)\min_{g^\top x=-1}\|Ax\|_\infty$
3: Initialize $x^{(0)} = \arg\min_{g^\top x=-1}\|Ax\|_2$; $L = \lfloor\log_{1+\varepsilon}\frac{\|Ax^{(0)}\|_2}{n^{1/2}}\rfloor$; $U = \lfloor\log_{1+\varepsilon}\|Ax^{(0)}\|_2\rfloor$
4: **while** $L < U$:
5:      $P = \lfloor\frac{L+U}{2}\rfloor$; $M = (1+\varepsilon)^P$
6:      **if** Subsolver$(A, g, \varepsilon, M)$ is infeasible **then** $L = P + 1$
7:      **else** let $x^{(t)}$ be the output of Subsolver$(A, g, \varepsilon, M)$; $U = P$, $t \leftarrow t+1$
8: **return** $x^{(t)}$

---

**Algorithm 2** $\ell_\infty$-regression Subsolver$(A, g, \varepsilon, M)$

---

1: **Output**: Find $x$ such that $\|Ax\|_\infty \leq (1+\varepsilon)M$ or find $r$ such that $\frac{\mathcal{E}(r)}{\|r\|_1} \geq \left(\frac{M}{1+\varepsilon}\right)^2$
2: **Initialize**: $r^{(0)} = w + \frac{d}{n}$, where $w$ is an $\ell_\infty$ Lewis weight overestimate vector of $A$
3: $t = 0$, $t' = 0$, $s^{(t')} = 0$
4: **while** $\|r^{(t)}\|_1 \leq \frac{\|r^{(0)}\|_1}{\varepsilon}$
5:      $x^{(t)} = \arg\min_{x:g^\top x=-1}\langle r^{(t)}, (Ax)^2\rangle$
6:      **if** $\frac{\langle r^{(t)}, (Ax^{(t)})^2\rangle}{\|r^{(t)}\|_1} \geq \left(\frac{M}{1+\varepsilon}\right)^2$ **then return** $r^{(t)}$
7:      **if** $\|Ax^{(t)}\|_\infty \leq (1+\varepsilon)M$ **then return** $x^{(t)}$
8:      **if** $\|Ax^{(t)}\|_\infty > S = d^{\frac{1}{3}}M$:                  ▷ *Case 1*
9:          Let $i$ be an index such that $|(Ax^{(t)})_i| = \|Ax^{(t)}\|_\infty$
10:          $r_j^{(t+1)} = \begin{cases} r_j^{(t)} & j \neq i \\ r_j^{(t)} + 1 & j = i \end{cases}$
11:      **else**:                                               ▷ *Case 2*
12:          Let $t' = t' + 1$; $s^{(t')} = s^{(t'-1)} + x^{(t)}$;
13:          **if** $\|As^{(t')}\|_\infty/t' \leq (1+\varepsilon)M$ **then return** $s^{(t')}/t'$
14:          $r_j^{(t+1)} = \begin{cases} r_j^{(t)}\frac{(Ax^{(t)})_j^2}{M^2} & \text{if } (Ax^{(t)})_j^2 \geq (1+\varepsilon)M^2 \\ r_j^{(t)} & \text{otherwise} \end{cases}$
15:      $t = t + 1$
16: **return** $r^{(t)}$

---

### 3.1 Algorithm

The main part of our algorithm is the subroutine shown in Algorithm 2 which takes as input a guess for the optimal objective of Problem 3. To obtain a $(1 + \varepsilon)$-approximation for this value, our main algorithm 1 proceeds by performing a binary search. Starting with an initial solution $x^{(0)} = \arg\min_{g^\top x=-1}\|Ax\|_2$, the algorithm performs a binary search on the $(1 + \varepsilon)$-grid between $\|Ax^{(0)}\|_2$ and $\frac{\|Ax^{(0)}\|_2}{n^{1/2}}$ (which are an upper bound and a lower bound for $\|Ax^*\|_\infty$). The number of guesses is at most $\log\frac{\log(\|Ax^{(0)}\|_\infty)-\log(\frac{\|Ax^{(0)}\|_2}{n^{1/2}})}{\varepsilon} = O(\log\frac{\log n}{\varepsilon})$.

Our main focus in this section is then to demonstrate that Algorithm 2, given a guess $M$, outputs a solution $x$ with $\|Ax\|_\infty \leq (1+\varepsilon)M$, if any. For our convenience, we start with the dual formulation of the problem with the squared objective

$$\min_{g^\top x=-1}\|Ax\|_\infty^2 = \min_{g^\top x=-1}\max_{\|r\|_1=1}\langle r, (Ax)^2\rangle = \max_{\|r\|_1=1}\min_{g^\top x=-1}\langle r, (Ax)^2\rangle = \max_{r\geq 0}\frac{\mathcal{E}(r)}{\|r\|_1},$$

where we use $\mathcal{E}(r)$ to denote the objective $\min_{g^\top x=-1}\langle r, (Ax)^2\rangle$. For a guess $M$ for the optimal objective of Problem 3, the goal of our algorithm is to produce a primal solution $x$ such that $g^\top x = -1$ and $x$ satisfies $\|Ax\|_\infty \leq (1+\varepsilon)M$ or certify that $\min_{g^\top x=-1}\|Ax\|_\infty^2 = \max_{r\geq 0}\frac{\mathcal{E}(r)}{\|r\|_1} \geq \left(\frac{M}{1+\varepsilon}\right)^2$, in which case we need to increase $M$.

The core idea of our algorithm is to maintain in each iteration $t$ the following invariant

$$\mathcal{E}(r^{(t+1)}) - \mathcal{E}(r^{(t)}) \geq M^2(\|r^{(t+1)}\|_1 - \|r^{(t)}\|_1). \tag{4}$$

The telescoping property of this invariant guarantees that if the algorithm outputs a dual solution $r^{(T)}$ with $\|r^{(T)}\|_1$ significantly large compared to the initial $\|r^{(0)}\|_1$, we will have $\frac{\mathcal{E}(r^{(T)})}{\|r^{(T)}\|_1} \geq \frac{M^2}{(1+\varepsilon)^2}$.

We leverage the energy increase lower bound lemmas A.2 and A.1 to determine the update for the dual solution $r$. First, the initial dual solution $r^{(0)}$ is set to $w + \frac{d}{n}$, where $w$ is a $\ell_\infty$ Lewis weight overestimate vector for matrix $A$, which can be efficiently obtained using the fixed point iteration from [CCLY19], which we revise in Section E of the appendix. The $\frac{d}{n}$ component is added to guarantee that the coordinates of the dual solution are not too small. With this initialization, subsequent update $r^{(t)}$ always satisfies the condition of Lemma A.2 and thus we can apply it when necessary. To update the dual solution, the novelty in our method is to distinguish between the two regimes. In the high width regime (Case 1), ie, when the corresponding minimizer $x^{(t)}$ to $\min_{g^\top x = -1} \langle r, (Ax)^2 \rangle$ satisfies $\|Ax^{(t)}\|_\infty > S$, where $S$ is set to $d^{\frac{1}{3}}M$, we update a single coordinate $i$ that achieves the maximum value $|(Ax^{(t)})_i| = \|Ax^{(t)}\|_\infty$ by setting $r_i^{(t+1)} = r_i^{(t)} + 1$. Lemma A.2 will guarantee that the invariant 4 holds, and at the same time, $\mathcal{E}\left(r^{(t)}\right)$ increases fast. In the low width regime (Case 2), when $\|Ax^{(t)}\|_\infty \leq S$, we use the lower bound given by Lemma A.1. In order to maintain the invariant A.2, we can guarantee for each coordinate $j$, $\dfrac{(Ax^{(t)})_j^2 r_j^{(t)} \left(1 - \frac{r_j^{(t)}}{r_j^{(t+1)}}\right)}{r_j^{(t+1)} - r_j^{(t)}} \geq M^2$. From here, we can derive the update rule as shown in Algorithm 2.

We give the full description in Algorithm 2.

### 3.2 Analysis

We show the full analysis of Algorithm 2 in Appendix B. To show Theorem 1.2, first, we can see that the algorithm performs $O\left(\log \frac{\log(\|Ax_0\|_\infty - \|Ax^*\|_\infty)}{\varepsilon}\right)$ binary search steps. Our analysis shows that, in each step, the algorithm uses $O\left(\left(\frac{1}{\varepsilon^2} + \frac{d^{1/3}}{\varepsilon}\right) \log \frac{n}{\varepsilon}\right)$ calls to the solver for problems of the form $\min_{x: g^\top x = -1} \langle r, (Ax)^2 \rangle$. By combining these two facts, we immediately have Theorem 1.2.

## 4 Quasi-Self-Concordant Optimization

---

**Algorithm 3** Algorithm for optimizing $h(x) = \sum_{i=1}^n f((Ax - b)_i)$

---

1: **Input**: initial solution $x_0$, lower bound $B$ on $h$, diameter $R$ of sub-level set $\mathcal{L}_0$.
2: **Output**: Find $x$ such that $h(x) \leq h(x^*) + \varepsilon$
3: **Initialize**: $x^{(0)} = x_0$
4: Let $T = O\left(CR \log\left(\frac{h(x^{(0)}) - B}{\varepsilon}\right)\right)$
5: **for** $(t = 0; t < T; t \leftarrow t + 1)$:  ▷ or terminate when $h(x^{(t+1)}) - h(x^{(t)}) \leq O(\varepsilon/CR)$
6:      $\boldsymbol{\Delta} \leftarrow \{\emptyset\}$  ▷ set of candidate updates
7:      **for** $\left(\nu = h(x^{(t)}) - B; \nu \geq \varepsilon; \nu \leftarrow \frac{1}{2}\nu\right):$  ▷ halving after each step
8:          **for** $\left(M = e^2\nu; M \geq \frac{\nu}{CR}; M \leftarrow \frac{1}{2}M\right):$  ▷ halving after each step
9:              Let $\Delta_{\nu,M}$ be a primal solution output by $\text{ResidualSolver}(x^{(t)}, M)$ if any
10:              $\boldsymbol{\Delta} \leftarrow \boldsymbol{\Delta} \cup \{\Delta_{\nu,M}\}$
11:          $x^{(t+1)} = x^{(t)} - \frac{1}{e^2}\Delta^{(t)}$ where $\Delta^{(t)} = \arg\min_{\Delta \in \boldsymbol{\Delta}} h\left(x^{(t)} - \frac{1}{e^2}\Delta\right)$  ▷ update step
12:

---

To simplify exposition, we present an algorithm for solving the *unconstrained* problem

$$\min_x h(x) := \sum_{i=1}^n f((Ax - b)_i) \tag{5}$$

---

**Algorithm 4** ResidualSolver$(x, M)$

---

1: **Initialize**: $r^{(0)} = w + \frac{d}{n}$, where $w$ is an $\ell_\infty$ Lewis weight overestimate vector of $A$

2: $g_i = \frac{-1}{M}(A^\top \nabla f(x))_i$, $s_i = f''((Ax)_i)$, $t = 0$, $t' = 0$, $v^{(t')} = 0$

3: **while** $\|r^{(t)}\|_1 \leq 2(\|w\|_1 + d)$

4:      Let $p^{(t)} = 2(\|w\|_1 + d)s + \frac{MC^2}{2}r^{(t)}$; $\Delta^{(t)} = \arg\min_{\Delta : g^\top \Delta = -1} \left\langle p^{(t)}, (A\Delta)^2 \right\rangle$

5:      **if** $\left\langle s + \frac{MC^2}{2}\frac{r^{(t)}}{\|r^{(t)}\|_1}, (A\Delta^{(t)})^2 \right\rangle \geq 13M$ **then return** $r^{(t)}$

6:      **if** $\|A\Delta^{(t)}\|_\infty \leq \frac{11}{C}$ **then return** $\Delta^{(t)}$

7:      **if** $\|A\Delta^{(t)}\|_\infty > S = \frac{11d^{\frac{1}{3}}}{C}$:                      ▷ *Case 1*

8:          Let $i$ be an index such that $|(A\Delta^{(t)})_i| = \|A\Delta^{(t)}\|_\infty$

9:          $r_j^{(t+1)} = \begin{cases} r_j^{(t)} & j \neq i \\ r_j^{(t)} + 1 & j = i \end{cases}$

10:      **else**:                                                     ▷ *Case 2*

11:          Let $t' = t' + 1$; $v^{(t')} = v^{(t'-1)} + \Delta^{(t)}$

12:          **if** $\|Av^{(t')}\|_\infty / t' \leq \frac{11}{C}$ **then return** $v^{(t')}/t'$

13:          $r_j^{(t+1)} = \begin{cases} \frac{1}{52}r_j^{(t)}(A\Delta^{(t)})_j^2 C^2 & \text{if } (A\Delta^{(t)})_j^2 \geq \frac{100}{C^2} \\ r_j^{(t)} & \text{otherwise} \end{cases}$

14:      $t = t + 1$

15: **return** $r^{(t)}$

---

where $f : \mathbb{R} \to \mathbb{R}$ is a $C$-quasi-self-concordant function defined in Definition 2.1, $A \in \mathbb{R}^{n \times d}$, and $b \in \mathbb{R}^n$ with $n \geq d$. In the appendix, we give an extension to the fully generalized constrained problem $\min_{x : Nx = v} \sum_{i=1}^n f((Ax - b)_i)$ as stated in Theorem 1.1.

**Notation:** We denote by $x^*$ the optimal solution to the problem and $x^{(0)}$ the initial solution used by our algorithm. Also, let us write $\nabla f(x) = (f'((Ax - b)_1), \ldots, f'((Ax - b)_n))^\top$, $\nabla^2 f(x) = \text{diag}(f''((Ax - b)_1), \ldots, f''((Ax - b)_n))$. We then have $\nabla h(x) = A^\top \nabla f(x)$ and $\nabla^2 h(x) = A^\top \nabla^2 f(x) A$.

**Assumptions:** Our algorithm and its analysis require the following assumptions.

**Assumption 1.** *We assume that there exists a finite value $B$ such that $h(x) \geq B$, for all $x \in \mathbb{R}^d$.*

**Assumption 2.** *Let $\mathcal{L}_0 = \{x : h(x) \leq h(x^{(0)})\}$. We assume that there exists a finite value $R$ such that $\max_{x \in \mathcal{L}_0} \|Ax - Ax^*\|_\infty \leq R$.*

We note that Assumption 1 is standard in prior work (eg. by [ABS21]). Many commonly used functions are non-negative and thus admit a trivial lower bound $B = 0$. Assumption 2 on the boundedness of the diameter of the level set is also common in prior work on trust-region Newton method, such as [KSJ18, Doi23]. Such dependencies are natural, and virtually all known algorithms for QSC optimization have a dependency on the size and shape of the trust region, as well as on the distance between the initial and the optimal point (e.g. [ABS21] define their distance parameter such that $\|Ax^*\|_\infty \leq R$ and $\|Ax^*\|_2 \leq R$).

**Our approach:** We follow a trust-region based template. In particular, our approach leverages the box-constrained Newton method [CMTV17], which was used in the context of matrix scaling, a special case of QSC minimization. A similar method was developed in parallel by [AZLOW17] and was employed by [ABS21] for optimizing problem 5. The general idea behind these trust region templates is that, if the Hessian of the objective is promised to not change by more than a constant multiplicative factor within an $\ell_\infty$ region centered around the current iterate, then minimizing the local quadratic approximation within this region yields a new iterate which makes significant progress in function value. In fact, this progress is comparable to the progress made by moving to the best point within the $\ell_\infty$ region. Hence the remaining challenge is to approximately solve the quadratic minimization problem subject to an $\ell_\infty$ constraint. Since the subproblem is very robust to approximation, it suffices to obtain a constant factor approximation to a regularized $\ell_\infty^2 + \ell_2^2$ problem, which we achieve via a new IRLS solver. Now we can provide formal statements.

Quasi-self-concordance implies Hessian stability, which allows us to reduce Problem 5 to solving a sequence of residual problems of the form

$$\max_{\|A\Delta\|_\infty \leq \frac{1}{C}} \operatorname{res}_x(\Delta) := \nabla f(x)^\top (A\Delta) - \frac{1}{e}(A\Delta)^\top \nabla^2 f(x)(A\Delta) \tag{6}$$

We prove that an approximate solution to Problem 6 allows us to make significant progress in the function value of $h$. The full proof can be found in Section C.

**Lemma 4.1.** *Consider an iteration $t$ of Algorithm 3, and let $x = x^{(t)}$ be the current iterate. Suppose the* $\operatorname{ResidualSolver}$ *can compute $\widetilde{\Delta}$ such that* $\operatorname{res}_x(\widetilde{\Delta}) \geq \kappa \max_{\|A\Delta\|_\infty \leq \frac{1}{C}} \operatorname{res}_x(\Delta)$. *Then we have*

$$h\left(x - \frac{\widetilde{\Delta}}{e^2}\right) - h(x^*) \leq \left(1 - \frac{\kappa}{e^2 CR}\right)(h(x) - h(x^*)).$$

*Consequently, Algorithm 3 constructs a solution $x^{(T)}$ such that $h(x^{(T)}) \leq h(x^*) + \varepsilon$ using $T = O\left(\frac{RC}{\kappa} \log \frac{h(x^{(0)}) - h(x^*)}{\varepsilon}\right)$ iterations of the outermost for loop.*

To solve Problem 6, we recast it in a more amenable form, which is related to an $\ell_\infty$-regression problem, but with the key difference that it contains an additional quadratic term $\langle s, (A\Delta)^2 \rangle$. Specifically, given a guess $M$ for the objective value, we define the minimization problem

$$\min_{g^\top \Delta = -1} \left\langle s, (A\Delta)^2 \right\rangle + \frac{MC^2}{2}\|A\Delta\|_\infty^2, \tag{7}$$

where $g_i = \frac{-1}{M}\left(A^\top \nabla f(x)\right)_i$ and $s_i = f''\left((Ax)_i\right)$ for all $i$. We show that algorithm $\operatorname{ResidualSolver}$ satisfies the following invariant.

**Invariant 1.** *The algorithm either outputs a solution $\Delta$ such that $g^\top \Delta = -1$, $\|A\Delta\|_\infty \leq \frac{11}{C}$, and $\left\langle s, (A\Delta)^2 \right\rangle \leq \min_{g^\top \Delta = -1} \left\langle s, (A\Delta)^2 \right\rangle + \frac{MC^2}{2}\|A\Delta\|_\infty^2$, or certifies that $\min_{g^\top \Delta = -1} \left\langle s, (A\Delta)^2 \right\rangle + \frac{MC^2}{2}\|A\Delta\|_\infty^2 \geq 13M$.*

The output of an algorithm satisfying Invariant 1 can be converted into a good approximate solution to Problem 6. We prove this formally in Lemma C.7. At this point the major challenge is providing such an algorithm. We show that we can achieve it by extending our approach for solving $\ell_\infty$-regression from Section 3. The key difference in this setting is the presence of an additional quadratic term, which makes the extension non-trivial and limits us to achieving only a constant-factor approximation. Nonetheless, this level of approximation suffices for our purposes.

The following lemma provides formal guarantees for the $\operatorname{ResidualSolver}$, and its full proof can be found in Section C.

**Lemma 4.2.** *For an iterate $x$, let $\Delta^* = \arg\max_{\|A\Delta\|_\infty \leq 1/C} \operatorname{res}_x(\Delta)$ and $\mathcal{OPT} = \operatorname{res}_x(\Delta^*)$. For $M$ such that $\mathcal{OPT} \in (\frac{M}{2}, M]$, Algorithm 4 outputs $\widehat{\Delta}$ such that $\|A\widehat{\Delta}\|_\infty \leq \frac{1}{C}$ and $\operatorname{res}_x(\widehat{\Delta}) \geq \frac{\mathcal{OPT}}{20}$, and makes $O(d^{1/3} \log n)$ calls to a linear system solver.*

Having provided the description of the $\operatorname{ResidualSolver}$ and its guarantee, we can now describe and analyze the main routine shown in Algorithm 3. The algorithm starts with a solution $x^{(0)}$ and iteratively updates it. In each iteration, it performs a binary search for the objective of residual problem 6 using the two for-loops in Line 7 and Line 8. With each guess for this objective, the algorithm uses the $\operatorname{ResidualSolver}$ (Algorithm 4) to find a solution. Algorithm 3 finally uses the best solution among the ones that are found to update the iterate. To analyze the algorithm, we first use the following fact from [ABS21].

**Lemma 4.3** (Lemma 4.3, 4.4 [ABS21]). *If $h(x^{(t)}) - h(x^*) \in (\frac{\nu}{2}, \nu]$, then $\mathcal{OPT} \in (\frac{\nu}{8CR}, e^2\nu]$. Further, if $\mathcal{OPT} \in (\frac{M}{2}, M]$ then $(A\Delta^*)^\top \nabla^2 f(x)(A\Delta^*) \leq eM$.*

This lemma guarantees that the steps executed in the two for-loops in Algorithm 3 are sufficient to find a sufficiently close guess for the residual problem objective. In total, the number of steps required to find $M$ such that $\mathcal{OPT} \in (\frac{M}{2}, M]$ is $O\left(\log(CR)\log\left(\frac{h(x^{(t)}) - B}{\varepsilon}\right)\right)$.

Finally, combining Lemma 4.1, Lemma 4.2 with the binary search procedure in Line 7-8 (Lemma 4.3), we can conclude that the total number iterations of Algorithm 3 to find an $\varepsilon$-additive solution is $O\left(CR\log(CR)\log^2\left(\frac{h(x^{(0)}) - h(x^*)}{\varepsilon}\right)\right)$, each of which uses $O\left(d^{1/3}\log n\right)$ linear system solves. This concludes the proof for Theorem 1.1.

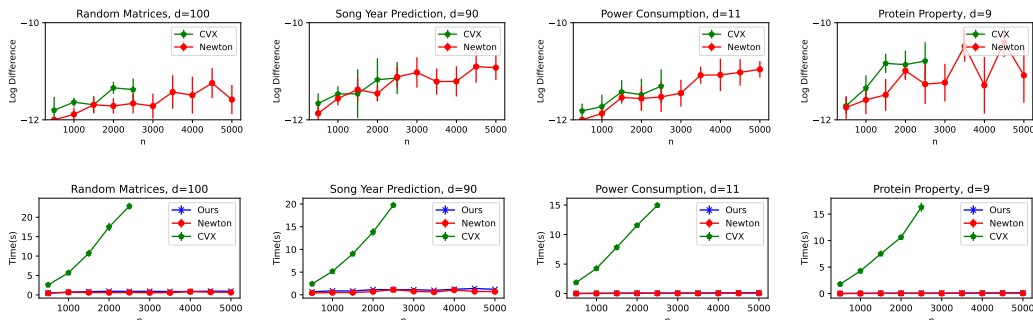

Figure 1: Runtime (in seconds) on random matrices and three real-world datasets when $p = 8$ and $\varepsilon = 10^{-10}$. We run each each experiment 5 times and report the mean and standard deviation of the runtime. The first row shows the absolute difference in the objectives returned by CVX and Newton's method against our algorithm on the $\log$ scale.

Table 1: Performance of our algorithm on the large instances. In the last row, we show the gap in the objective value between Newton's method and our algorithm.

| Dataset | Random Matrices | Song Year Prediction [BM11] | Consumption of Power [HB06] | Protein Property [Ran13] |
|---|---|---|---|---|
| Size | $100000 \times 100$ | $463811 \times 90$ | $1844352 \times 11$ | $41157 \times 9$ |
| Newton (s) | 0.5 | 2.7 | 3.0 | 0.2 |
| Ours (s) | 1.3 | 7.5 | 6.5 | 0.3 |
| Gap | $-3.6 \times 10^{-12}$ | $2.6 \times 10^{-9}$ | $4.5 \times 10^{-9}$ | $-7.1 \times 10^{-11}$ |

## 5 Experiments

We test our algorithm on $\ell_2$-regularized $\ell_p$-regression problems: $\min_x \|Ax - b\|_p^p + \mu \|Ax - b\|_2^2$. For $p \geq 3$, the function $|x|^p + \mu x^2$ is $C$-QSC for $C = p\mu^{-1/(p-2)}$.

**Baseline**. We compare our algorithm runtime against CVX and the Newton's method in [KSJ18]. For the Newton's method, we use a line search to set the step size in each iteration.

**Problems and datasets**. We study the $\ell_p + \ell_2$ regression problem in two settings: 1) Random matrix $A, b$: The entries of $A, b$ are generated uniformly at random between 0 and 1, the second dimension ($d$) of $A$ is fixed to 100; 2) On real world datasets: we use for real-world datasets available for regression tasks on UCI repository. The dataset sizes are reported in Table 1. Since CVX slows down significantly on larger instances, we only randomly pick up to 2500 in each instance. For our algorithm and the Newton's method, we also measure the time they run on larger instances and the entire datasets. In all experiments, we use $p = 8$ and precision $\varepsilon = 10^{-10}$, $\mu = 1$.

**Correctness**. To verify the correctness of our algorithms, we assume that CVX gives a solution with high precision. We plot the absolute difference between the objectives outputted by CVX and Newton's method and the objective by our algorithm. In all cases, the difference is within the margin of error ($\varepsilon = 10^{-10}$), indicating that our algorithm outputs high-precision solutions.

**Performance**. Implementations were done on MATLAB 2024a on a MacBook Pro M2/16GB RAM. The performance of all three algorithms is reported in Figure 1. On all small instances, our algorithm and the Newton's method have comparable runtime and are both significantly faster than CVX.

We further compare the performance of our algorithm and the Newton's on large instances (reported in Table 1). Both algorithms are efficient, and are even faster than the time CVX runs on small instances. However, there is a gap in the runtime between our algorithm and Newton's method. We note that Newton's method has been shown to perform very well in practice. Nevertheless, it comes with a significantly weaker theoretical guarantee [BV04]. Our algorithm, on the other hand, provides a stronger theoretical convergence rate and, as a proof of concept, gives a close comparable performance.

## Acknowledgement

AE was supported in part by an Alfred P. Sloan Research Fellowship. AV was supported by the French Agence Nationale de la Recherche (ANR) under grant ANR-21-CE48-0016 (project COMCOPT).

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

## A Energy Lemmas

**Lemma A.1.** *Given $r, r' \in \mathbb{R}^n$ and $r' \geq r$. Let $x = \arg\min_{x:g^\top x=-1} \langle r, (Ax)^2 \rangle$, then*

$$\mathcal{E}(r') - \mathcal{E}(r) \geq \sum_{i=1}^n (Ax)_i^2 r_i \left(1 - \frac{r_i}{r_i'}\right).$$

The proof for this lemma can be found in prior works such as [EV19] (for a special case) and [ABS21] (Lemma C.1).

Via $\ell_\infty$ Lewis weight overestimates, [JLS22] also show the following

**Lemma A.2** (Lemma 3.6 [JLS22]). *Let $w$ be an $\ell_\infty$ Lewis weight overestimates vector for matrix $A$. For $r \geq w$, let $x = \arg\min_{x:g^\top x=-1} \langle r^{(t)}, (Ax)^2 \rangle$. Then for $v \in \mathbb{R}_{\geq 0}^n$ such that $\|v\|_1 \leq 1$*

$$\mathcal{E}(r + v) - \mathcal{E}(r) \geq \frac{1}{2} \sum v_i (Ax)_i^2.$$

## B Analysis of the $\ell_\infty$-Regression Algorithm

In this section, we give the analysis of the main subroutine of our $\ell_\infty$-regression algorithm, shown in Algorithm 2 .

### B.1 Correctness

Now we will show the correctness of our algorithm. That is, Algorithm 2 produces a primal solution $x$ such that $g^\top x = -1$ and $x$ satisfies $\|Ax\|_\infty \leq (1 + \varepsilon)M$ or a dual solution $r$ with $\frac{\mathcal{E}(r)}{\|r\|_1} \geq \left(\frac{M}{1+\varepsilon}\right)^2$.

First, we can see that if Algorithm 2 outputs a primal solution $x$ in Line 10 and Line 17, we immediately have $\|Ax\|_\infty \leq (1 + \varepsilon)M$. Therefore, we only have to show guarantees for the dual solution. As hinted, we first show that Algorithm 2 maintains invariant 4 in the following lemma.

**Lemma B.1.** *Algorithm 2 maintains that for all $t \geq 0$,*

$$\frac{\mathcal{E}(r^{(t+1)}) - \mathcal{E}(r^{(t)})}{\|r^{(t+1)}\|_1 - \|r^{(t)}\|_1} \geq M^2.$$

*Proof.* When $\|Ax\|_\infty \geq S$, we have $r_j^{(t+1)} = \begin{cases} r_j^{(t)} & j \neq i \\ r_j^{(t)} + 1 & j = i \end{cases}$, for a coordinate $i$ such that $|(Ax^{(t)})_i| = \|Ax^{(t)}\|_\infty$. Let $v = \mathbb{1}_i$ be the indicator vector for coordinate $i$. We have $r^{(t+1)} = r^{(t)} + v$ and $\|v\|_1 = 1$. Since $r^{(t)} \geq r^{(0)} > w$, we can apply Lemma A.2 and have

$$\frac{\mathcal{E}(r^{(t+1)}) - \mathcal{E}(r^{(t)})}{\|r^{(t+1)}\|_1 - \|r^{(t)}\|_1} \geq \frac{\frac{1}{2} v_i (Ax^{(t)})_i^2}{v_i} \geq \frac{S^2}{2} = \frac{d^{2/3}M^2}{2} \geq M^2.$$

When $\|Ax\|_\infty < S$, for each coordinate $j$ such that $r_j^{(t+1)} > r_j^{(t)}$ we have $r_j^{(t+1)} = r_j^{(t)} \frac{(Ax^{(t)})_j^2}{M^2}$. Hence,

$$\frac{(Ax^{(t)})_j^2 r_j^{(t)} \left(1 - \frac{r_j^{(t)}}{r_j^{(t+1)}}\right)}{r_j^{(t+1)} - r_j^{(t)}} = \frac{(Ax^{(t)})_j^2 r_j^{(t)}}{r_j^{(t+1)}} = M^2.$$

By Lemma A.1,

$$\frac{\mathcal{E}(r^{(t+1)}) - \mathcal{E}(r^{(t)})}{\|r^{(t+1)}\|_1 - \|r^{(t)}\|_1} \geq \frac{\sum_j (Ax^{(t)})_j^2 r_j^{(t)} \left(1 - \frac{r_j^{(t)}}{r_j^{(t+1)}}\right)}{\sum_j r_j^{(t+1)} - r_j^{(t)}} \geq M^2.$$

$\square$

Lemma B.1 allows us to show the quality of the dual solution, if output.

**Lemma B.2.** *If Algorithm 2 returns a dual solution $r^{(T)}$ then $\frac{\mathcal{E}(r^{(T)})}{\|r^{(T)}\|_1} \geq \left(\frac{M}{1+\varepsilon}\right)^2$.*

*Proof.* The case when the Algorithm terminates in Line 8 immediately gives us the claim. We only have to care about the case when the while-loop terminates with $\|r^{(T)}\|_1 > \frac{\|r^{(0)}\|_1}{\varepsilon}$. By Lemma B.1, we have that

$$\sum_{i=0}^{T-1} \mathcal{E}(r^{(t+1)}) - \mathcal{E}(r^{(t)}) \geq M^2 \sum_{i=0}^{T-1} \|r^{(t+1)}\|_1 - \|r^{(t)}\|_1$$

leading to $\mathcal{E}(r^{(T)}) \geq M^2(\|r^{(T)}\|_1 - \|r^{(0)}\|_1)$. Since $\|r^{(T)}\|_1 > \frac{\|r^{(0)}\|_1}{\varepsilon}$

$$\frac{\mathcal{E}(r^{(T)})}{\|r^{(T)}\|_1} \geq M^2 \left(1 - \frac{\|r^{(0)}\|_1}{\|r^{(T)}\|_1}\right) \geq M^2 (1 - \varepsilon) \geq \left(\frac{M}{1+\varepsilon}\right)^2,$$

as needed. $\square$

### B.2 Runtime

To bound the runtime of Algorithm 2, we also splits the iteration by the width of $\|Ax^{(t)}\|_\infty$. Let $T_{hi}$ and $T_{lo}$ be the iterations when $\|Ax^{(t)}\|_\infty > S$ and $\|Ax^{(t)}\|_\infty \leq S$, respectively before the algorithm returns a solution. We also abuse the notations and denote by $T_{hi}$ and $T_{lo}$ the numbers of such iterations, respectively. The following lemmas bound $T_{hi}$ and $T_{lo}$.

**Lemma B.3.** $T_{hi} \leq \frac{6d^{1/3}}{\varepsilon}$.

*Proof.* Again, for $t$ such that $\|Ax^{(t)}\|_\infty > S$, by Lemma A.2, we have

$$\frac{\mathcal{E}(r^{(t+1)}) - \mathcal{E}(r^{(t)})}{\|r^{(t+1)}\|_1 - \|r^{(t)}\|_1} \geq \frac{\frac{1}{2}v_i(Ax^{(t)})_i^2}{v_i} \geq \frac{S^2}{2} = \frac{d^{2/3}M^2}{2}.$$

Since $\|r^{(t+1)}\|_1 - \|r^{(t)}\|_1 = 1$, we get $\mathcal{E}(r^{(t+1)}) - \mathcal{E}(r^{(t)}) \geq \frac{d^{2/3}M^2}{2}$. Therefore $\mathcal{E}(r^{(T)}) \geq T_{hi} \cdot \frac{d^{2/3}M^2}{2}$. Notice that $\|r^{(T)}\|_1 \leq \frac{\|r^{(0)}\|_1}{\varepsilon} \leq \frac{3d}{\varepsilon}$ and $\frac{\mathcal{E}(r^{(T)})}{\|r^{(T)}\|_1} \leq \left(\frac{M}{1+\varepsilon}\right)^2$, we get that $T_{hi} \leq \frac{6d^{1/3}}{\varepsilon}$. $\square$

**Lemma B.4.** $T_{lo} \leq O\left(\left(\frac{1}{\varepsilon^2} + \frac{d^{\frac{1}{3}}}{\varepsilon \log d}\right) \log \frac{n}{\varepsilon}\right)$.

*Proof.* Let $\alpha_j^{(t)} = \frac{r_j^{(t+1)}}{r_j^{(t)}}$. Before the algorithm terminates, we have

$$\left\| A \frac{\sum_{t \in T_{lo}} x^{(t)}}{T_{lo}} \right\|_\infty \geq (1+\varepsilon)M$$

This means there must exist a coordinate $i$ such that $\sum_{t \in T_{lo}} \frac{|(Ax^{(t)})_i|}{M} \geq (1 + \varepsilon)T_{lo}$. We have if $\frac{(Ax^{(t)})_i^2}{M^2} < (1 + \varepsilon)$, then $\frac{|(Ax^{(t)})_i|}{M} < (1 + \frac{\varepsilon}{2})$, and if $\frac{(Ax^{(t)})_i^2}{M^2} \geq (1 + \varepsilon)$, $\frac{(Ax^{(t)})_i^2}{M^2} = \alpha_i^{(t)} > 1$. Hence,

$$\frac{|(Ax^{(t)})_i|}{M} \leq (1 + \frac{\varepsilon}{2}) + \mathbb{1}_{\alpha_i^{(t)} > 1} \sqrt{\alpha_i^{(t)}}$$

We obtain

$$T_{lo}(1 + \frac{\varepsilon}{2}) + \sum_{t \in T_{lo}, \alpha_i^{(t)} > 1} \sqrt{\alpha_i^{(t)}} \geq (1 + \varepsilon)T_{lo}$$

and thus $\sum_{t \in T_{lo}, \alpha_i^{(t)} > 1} \sqrt{\alpha_i^{(t)}} \geq \frac{\varepsilon T_{lo}}{2}$. For $t$ such that $\alpha_i^{(t)} > 1$, we have $\sqrt{\alpha_i^{(t)}} = \frac{|(Ax^{(t)})_i|}{M} \in [(1 + \varepsilon)^{\frac{1}{2}}, d^{\frac{1}{3}}]$. By Lemma A.1 from [EV19], we have a lower bound for $\prod_{t \in T_{lo}, \alpha_i^{(t)} > 1} \alpha_i^{(t)}$:

$$\prod_{t \in T_{lo}, \alpha_i^{(t)} > 1} \alpha_i^{(t)} \geq \min \left\{ (1 + \varepsilon)^{\frac{\varepsilon T_{lo}}{2(1+\varepsilon)^{\frac{1}{2}}}}, d^{\frac{2}{3} \frac{\varepsilon T_{lo}}{2d^{\frac{1}{3}}}} \right\} \tag{8}$$

On the other hand, $\frac{r_i^{(T)}}{r_i^{(0)}}$ is an upperbound for $\prod_{t \in T_{lo}, \alpha_i^{(t)} > 1} \alpha_i^{(t)}$. We initialize $r_i^{(0)} \geq \frac{d}{n}$ and before the algorithm terminates $r_i^{(T)} \leq \|r^{(T)}\|_1 \leq \frac{w+d}{\varepsilon} \leq \frac{3d}{\varepsilon}$. We now have that

$$\prod_{t \in T_{lo}, \alpha_i^{(t)} > 1} \alpha_i^{(t)} \leq \frac{r_i^{(T)}}{r_i^{(0)}} \leq \frac{3d}{\varepsilon} \cdot \frac{n}{d} \leq \frac{3n}{\varepsilon} \tag{9}$$

From (8) and (9), we obtain $T_{lo} \leq O\left(\left(\frac{1}{\varepsilon^2} + \frac{d^{\frac{1}{3}}}{\varepsilon \log d}\right) \log \frac{3n}{\varepsilon}\right)$. $\qquad \square$

## C  Analysis of the QSC Algorithm

In this section, we analyze our main Algorithm 3 for QSC optimization. The algorithm starts with a solution $x^{(0)}$ and iteratively updates it. Building on prior work [CMTV17, AZLOW17, ABS21], we show that the algorithm returns a nearly-optimal solution provided we have access to an algorithm for solving the following residual problems. Letting $x^{(t)}$ be the solution in iteration $t$ of Algorithm 3, we would like to find an approximate solution to the following residual problem:

$$\max_{\Delta : \|A\Delta\|_\infty \leq \frac{1}{C}} \mathrm{res}_{x^{(t)}}(\Delta) \coloneqq \nabla f(x^{(t)})^\top (A\Delta) - \frac{1}{e}(A\Delta)^\top \nabla^2 f(x^{(t)})(A\Delta) \tag{10}$$

Using a binary search approach as in [ABS21], we can find a value $M$ that is a 2-approximation to the optimal value of the above residual problem. This binary search is shown in the two for-loops in Line 7 and Line 8 of Algorithm 3. With each guess $M$, the algorithm uses the ResidualSolver (Algorithm 4) to find a solution to the residual problem. We show that if the residual solver returns a constant factor approximation to the residual problem then Algorithm 3 returns a nearly-optimal solution. We now give the formal analysis.

We start by showing that the steps executed in the two for-loops in Algorithm 3 are sufficient to find a sufficiently close guess for the residual problem objective. The lemma follows from Lemmas 4.3, 4.4 in [ABS21].

**Lemma C.1** (Lemma 4.3, 4.4 [ABS21]). *For each value $t$ of the outer-most iteration with iterate $x^{(t)}$, there is an inner-most iteration of Algorithm 3 for which the following hold. Let $M$ be the value considered in that iteration. Consider the residual problem:*

$$\max_{\Delta : \|A\Delta\|_\infty \leq 1/C} \mathrm{res}_{x^{(t)}}(\Delta) = \nabla f(x^{(t)})^\top (A\Delta) - \frac{1}{e}(A\Delta)^\top \nabla^2 f(x^{(t)})(A\Delta)$$

*Let $\Delta^* \in \arg\max_{\|A\Delta\|_\infty \leq 1/C} \mathrm{res}_{x^{(t)}}(\Delta)$ be an optimal solution to the residual problem, and let $\mathcal{OPT} = \mathrm{res}_{x^{(t)}}(\Delta^*)$ be its objective value. We have $\mathcal{OPT} \in (\frac{M}{2}, M]$ and $(A\Delta^*)^\top \nabla^2 f(x^{(t)})(A\Delta^*) \leq eM$.*

Next, we show that $\mathrm{ResidualSolver}(x^{(t)}, M)$ returns a constant factor approximation to the residual problem $\max_{\Delta:\ \|A\Delta\|_\infty \le 1/C} \mathrm{res}_{x^{(t)}}(\Delta)$ when we run it with the value $M$ guaranteed by the above lemma. We will provide the proof of this lemma in Section C.1.

**Lemma C.2.** *Consider an outermost iteration $t$ of Algorithm 3 and an innermost iteration with a value $M$ with the properties guaranteed by Lemma C.1. Let $\widetilde{\Delta} := \Delta_{\nu,M}$ be the solution returned by* $\mathrm{ResidualSolver}(x^{(t)}, M)$. *We have $\|A\widetilde{\Delta}\|_\infty \le \frac{1}{C}$ and $\mathrm{res}_{x^{(t)}}(\widetilde{\Delta}) \ge \frac{\mathcal{OPT}}{20}$.*

Equipped with the above guarantee, we can show that the algorithm converges in a small number of iterations. To this end, we first prove the following lemma which shows that each iteration significantly reduces the optimality gap.

**Lemma C.3.** *Let $x$ be an iterate. Suppose that $\mathrm{ResidualSolver}(x, M)$ returns a solution $\widetilde{\Delta}$ satisfying $\|A\widetilde{\Delta}\|_\infty \le \frac{1}{C}$ and $\mathrm{res}_x(\widetilde{\Delta}) \ge \kappa \max_{\|A\Delta\|_\infty \le \frac{1}{C}} \mathrm{res}_x(\Delta)$. We have*

$$h\left(x - \frac{\widetilde{\Delta}}{e^2}\right) - h(x^*) \le \left(1 - \frac{\kappa}{e^2 CR}\right)(h(x) - h(x^*)).$$

In order to prove Lemma C.3, first we recall the notion of Hessian stability.

**Definition C.1.** A function $h : \mathbb{R}^d \to \mathbb{R}$ is $(r, d(r))$-Hessian stable in the $\ell_\infty$-norm iff for all $x, y$ such that $\|x - y\|_\infty \le r$ we have

$$\frac{1}{d(r)} \nabla^2 h(x) \preceq \nabla^2 h(y) \preceq d(r) \nabla^2 h(x).$$

**Fact C.1** (From [CJJ$^+$20]). *For a $C$-quasi-self-concordant function $f$, $\sum_i f(x_i)$ is $\left(r, e^{Cr}\right)$-Hessian stable in the $\ell_\infty$-norm.*

We now give the proof of Lemma C.3.

*Proof.* Let $x^* = \arg\min h(x)$, we have $\|Ax - Ax^*\|_\infty \le R$, and $h$ is $(r, e)$-Hessian stable where $r = \frac{1}{C}$. Let $\widehat{x} = \frac{r}{R} x^* + \left(1 - \frac{r}{R}\right) x$; and let $\widehat{\Delta} = x - \widehat{x} = \frac{r}{R}(x - x^*)$. We have $\left\|A\widehat{\Delta}\right\|_\infty \le r$. Suppose that $\Delta^*$ is an optimal solution to the residual problem. Also recall the notation $\nabla h(x) = A^\top \nabla f(x)$ and $\nabla^2 h(x) = A^\top \nabla^2 f(x) A$ where we write $\nabla f(x) = \begin{pmatrix} f'((Ax - b)_1) \\ \cdots \\ f'((Ax - b)_n) \end{pmatrix}$, $\nabla^2 f(x) = \mathrm{diag}(f''((Ax - b)_1), \ldots, f''((Ax - b)_n))$.

Let $k = e^2$ and using hessian stability, we have

$$h\left(x - \widehat{\Delta}\right) - h(x) \ge -\nabla f(x)^\top \left(A\widehat{\Delta}\right) + \frac{1}{\exp(Cr)} \left(A\widehat{\Delta}\right)^\top \nabla^2 f(x) \left(A\widehat{\Delta}\right)$$

$$= -\nabla f(x)^\top \left(A\widehat{\Delta}\right) + \frac{1}{e} \left(A\widehat{\Delta}\right)^\top \nabla^2 f(x) \left(A\widehat{\Delta}\right)$$

$$= -\mathrm{res}_x\left(\widehat{\Delta}\right);$$

$$h\left(x - \frac{\widetilde{\Delta}}{k}\right) - h(x) \le -\nabla f(x)^\top \left(\frac{A\widetilde{\Delta}}{k}\right) + \exp\left(\frac{Cr}{k}\right) \left(\frac{A\widetilde{\Delta}}{k}\right)^\top \nabla^2 g(x) \left(\frac{A\widetilde{\Delta}}{k}\right)$$

$$\le \frac{1}{k}\left(-\nabla f(x)^\top \left(A\widetilde{\Delta}\right) + \frac{1}{e} \left(A\widetilde{\Delta}\right)^\top \nabla^2 f(x) \left(A\widetilde{\Delta}\right)\right)$$

$$= -\frac{1}{k} \mathrm{res}_x\left(\widetilde{\Delta}\right).$$

Since $\mathrm{res}\left(\widetilde{\Delta}\right) \geq \kappa\mathrm{res}_x\left(\Delta^*\right) \geq \kappa\mathrm{res}_x\left(\widehat{\Delta}\right)$, we have

$$h\left(x - \frac{\widetilde{\Delta}}{k}\right) - h\left(x\right) \leq -\frac{\kappa}{k}\mathrm{res}_x\left(\widehat{\Delta}\right)$$

$$\leq \frac{\kappa}{k}\left(h\left(x - \widehat{\Delta}\right) - h\left(x\right)\right)$$

$$= \frac{\kappa}{k}\left(h\left(\widehat{x}\right) - h\left(x\right)\right)$$

$$\leq \frac{\kappa r}{kR}\left(h\left(x^*\right) - h\left(x\right)\right)$$

This give us the conclusion

$$h\left(x - \frac{\widetilde{\Delta}}{k}\right) - h\left(x^*\right) \leq \left(1 - \frac{\kappa r}{kR}\right)\left(h\left(x\right) - h\left(x^*\right)\right).$$

$\square$

By combining Lemmas C.2 and C.3, we obtain the following convergence guarantee.

**Lemma C.4.** *Algorithm 3 constructs a solution $x^{(T)}$ such that $h(x^{(T)}) \leq h(x^*) + \varepsilon$ using $T = O\left(RC\log\left(\frac{h(x^{(0)}) - h(x^*)}{\varepsilon}\right)\right)$ iterations of the outermost loop.*

Next, we analyze the overall running time of Algorithm 3. We have the following upper bound on number of calls that Algorithm 3 makes to the residual solver.

**Lemma C.5.** *In each iteration $t$, Algorithm 3 makes $O\left(\log\left(CR\right)\log\left(\frac{h(x^{(0)}) - h(x^*)}{\varepsilon}\right)\right)$ calls to the residual solver 4.*

*Proof.* The algorithm executes a binary search using two for-loops 7 and 8. The number of steps of for-loop 7 is $O\left(\log\frac{h(x^{(t)}) - B}{\varepsilon}\right) = O\left(\log\frac{h(x^{(0)}) - h(x^*)}{\varepsilon}\right)$ and the number of steps of for-loop 8 is $O\left(\log CR\right)$, giving us the claim. $\square$

In Section C.6, we show the following upper bound on the running time of the ResidualSolver.

**Lemma C.6.** ResidualSolver *uses $O(d^{1/3}\log n)$ linear system solves.*

Finally, to conclude the proof of Theorem 1.1, we only have to combine Lemmas C.4, C.5, C.6.

## C.1 Proof of Lemma C.2

In this section, we prove the approximation guarantee of the ResidualSolver stated in Lemma C.2. Let $x$ and $M$ be the input to the ResidualSolver. As before, we let $\Delta^* \in \max_{\Delta:\ \|A\Delta\|_\infty \leq 1/C}\mathrm{res}_x(\Delta)$ and $\mathcal{OPT} = \mathrm{res}_x(\Delta^*)$. As in Section 4, we let $g_i = \frac{-1}{M}\left(A^\top\nabla f(x)\right)_i$ and $s_i = f''\left((Ax)_i\right)$ for all $i$.

We start with the following lemma.

**Lemma C.7.** *Consider an iterate $x$ and a guess $M$. Consider an algorithm that takes as input $x$ and $M$ and it outputs a solution to the Problem 7. If $\mathcal{OPT} \in (\frac{M}{2}, M]$ and the algorithm satisfies Invariant 1, then the algorithm outputs a solution $\Delta$ such that for $\widehat{\Delta} = \frac{\Delta}{11}$, we have $\|A\widehat{\Delta}\|_\infty \leq \frac{1}{C}$ and $\mathrm{res}_x(\widehat{\Delta}) \geq \frac{\mathcal{OPT}}{20}$.*

*Proof.* Let $a = -g^\top\Delta^*$. Note that $\frac{\Delta^*}{a}$ satisfies $g^\top\frac{\Delta^*}{a} = -1$. Since $\mathcal{OPT} \in (\frac{M}{2}, M]$, we have

$$a = \frac{\nabla f\left(x\right)^\top\left(A\Delta^*\right)}{M} \geq \frac{\mathcal{OPT}}{M} \geq \frac{1}{2}.$$

Moreover since $\|A\Delta^*\|_\infty \le \frac{1}{C}$ and from Lemma 4.3, we have

$$\left\langle s, \left(A\frac{\Delta^*}{a}\right)^2 \right\rangle + \frac{MC^2}{2}\left\|A\frac{\Delta^*}{a}\right\|_\infty^2 \le 4eM + 2M < 13M.$$

This means, our algorithm will output a solution $\Delta$ that satisfies

$$\nabla f(x)^\top (A\Delta) = M$$

$$\|A\Delta\|_\infty \le \frac{11}{C}$$

$$\left\langle s, (A\Delta)^2 \right\rangle \le \min_{g^\top \Delta = -1}\left(\left\langle s, (A\Delta)^2\right\rangle + \frac{MC^2}{2}\|A\Delta\|_\infty^2\right) < 13M$$

Let $\widehat{\Delta} = \frac{\Delta}{11}$, we have

$$\nabla f(x)^\top \left(A\widehat{\Delta}\right) = \frac{M}{11}$$

$$\left\|A\widehat{\Delta}\right\|_\infty \le \frac{1}{C}$$

$$\left\langle s, \left(A\widehat{\Delta}\right)^2\right\rangle < \frac{13}{121}M$$

which gives us

$$\mathrm{res}_x\left(\widehat{\Delta}\right) \ge \frac{M}{11} - \frac{1}{e}\cdot\frac{13}{121}M > \frac{M}{20} \ge \frac{\mathcal{OPT}}{20}.$$

$\square$

Thus it only remains to show that ResidualSolver satisfies the aforementioned invariant. Notice that the Problem 7 has a similar structure as an $\ell_\infty$-regression problem solved in the previous section, albeit with an additional quadratic term $\left\langle s, (A\Delta)^2\right\rangle$. Fortunately, the algorithm only needs to return a constant factor approximation, instead of a $1 + \varepsilon$ approximation. This allows us to extend the approach we used for Algorithm 2, and use a similar analysis.

In the remainder of this section, we show the following lemma, using similar ideas as in our analysis of our $\ell_\infty$ regression algorithm.

**Lemma C.8.** *Algorithm 4 satisfies Invariant 1.*

By combining Lemmas C.7 and C.8, we obtain Lemma 4.2.

We proceed with the proof Lemma C.8 by showing the following lemmas.

**Lemma C.9.** *For all iterations $t \ge 1$ of* ResidualSolver*, we have*

$$\left\langle s, \left(A\Delta^{(t)}\right)^2\right\rangle \le \min_{g^\top\Delta=-1}\left\langle s, (A\Delta)^2\right\rangle + \frac{MC^2}{2}\|A\Delta\|_\infty^2.$$

*Proof.* Let $\Delta^* = \arg\min_{g^\top\Delta=-1}\left\langle s, (A\Delta)^2\right\rangle + \frac{MC^2}{2}\|A\Delta\|_\infty^2$. We have

$$2\left(\|w\|_1 + d\right)\left\langle s, \left(A\Delta^{(t)}\right)^2\right\rangle$$

$$\le \left\langle \frac{MC^2}{2}r^{(t)} + 2\left(\|w\|_1 + d\right)s, \left(A\Delta^{(t)}\right)\right\rangle$$

$$\le \left\langle \frac{MC^2}{2}r^{(t)} + 2\left(\|w\|_1 + d\right)s, (A\Delta^*)\right\rangle \qquad \text{due to the optimality of } x^{(t)}$$

$$\le 2\left(\|w\|_1 + d\right)\left\langle \frac{MC^2}{2}\frac{r^{(t)}}{\|r^{(t)}\|_1} + s, (A\Delta^*)^2\right\rangle \qquad \text{since } \left\|r^{(t)}\right\|_1 \le 2\left(\|w\|_1 + d\right)$$

which gives

$$\left\langle s, \left(A\Delta^{(t)}\right)^2 \right\rangle \leq \left\langle \frac{MC^2}{2} \frac{r^{(t)}}{\left\|r^{(t)}\right\|_1} + s, (A\Delta^*)^2 \right\rangle$$

$$\leq \left\langle s, (A\Delta^*)^2 \right\rangle + \frac{MC^2}{2} \left\|A\Delta^*\right\|_\infty^2 ,$$

as needed. $\qquad\square$

**Lemma C.10.** ResidualSolver *maintains the invariant that, for all $t \geq 0$,*

$$\frac{\mathcal{E}\left(p^{(t+1)}\right) - \mathcal{E}\left(p^{(t)}\right)}{\left\|r^{(t+1)}\right\|_1 - \left\|r^{(t)}\right\|_1} \geq 26M.$$

*Proof.* When $\left\|A\Delta^{(t)}\right\|_\infty \geq S = \frac{11d^{\frac{1}{3}}}{C}$, we have $r_j^{(t+1)} = \begin{cases} r_j^{(t)} & j \neq i \\ r_j^{(t)} + 1 & j = i \end{cases}$, for a coordinate $i$ such that $\left(A\Delta^{(t)}\right)_i = \left\|A\Delta^{(t)}\right\|_\infty$. Let $v = \mathbb{1}_i$ be the indicator vector for coordinate $i$. We have $r^{(t+1)} = r^{(t)} + v$ and $\|v\|_1 = 1$. Using Lemma A.2 we have

$$\frac{\mathcal{E}\left(p^{(t+1)}\right) - \mathcal{E}\left(p^{(t)}\right)}{\left\|r^{(t+1)}\right\|_1 - \left\|r^{(t)}\right\|_1} \geq \frac{\frac{MC^2}{2} \cdot \frac{1}{2} v_i \left(A\Delta^{(t)}\right)_i^2}{v_i} \geq \frac{MC^2 S^2}{4} = 26Md^{\frac{2}{3}} \geq 26M.$$

When $\left\|A\Delta^{(t)}\right\|_\infty < S$, for each coordinate $j$ such that $r_j^{(t+1)} > r_j^{(t)}$ we have $r_j^{(t+1)} = \frac{1}{52} r_j^{(t)} \left(A\Delta^{(t)}\right)_j^2 C^2$. Note that $\frac{p_j^{(t)}}{p^{(t+1)}} \geq \frac{r_j^{(t)}}{r_j^{(t+1)}}$. Hence,

$$\frac{\left(A\Delta^{(t)}\right)_j^2 p_j^{(t)} \left(1 - \frac{p_j^{(t)}}{p_j^{(t+1)}}\right)}{r_j^{(t+1)} - r_j^{(t)}} = \frac{\left(A\Delta^{(t)}\right)_j^2 \frac{p_j^{(t)}}{p_j^{(t+1)}} \left(p_j^{(t+1)} - p_j^{(t)}\right)}{r_j^{(t+1)} - r_j^{(t)}}$$

$$\geq \frac{\frac{MC^2}{2} \left(A\Delta^{(t)}\right)_j^2 \frac{r_j^{(t)}}{r_j^{(t+1)}} \left(r_j^{(t+1)} - r_j^{(t)}\right)}{r_j^{(t+1)} - r_j^{(t)}}$$

$$= 26M.$$

by Lemma A.1,

$$\frac{\mathcal{E}\left(p^{(t+1)}\right) - \mathcal{E}\left(p^{(t)}\right)}{\left\|r^{(t+1)}\right\|_1 - \left\|r^{(t)}\right\|_1} \geq \frac{\sum_j \left(A\Delta^{(t)}\right)_j^2 p_j^{(t)} \left(1 - \frac{p_j^{(t)}}{p_j^{(t+1)}}\right)}{\sum_j \left(r_j^{(t+1)} - r_j^{(t)}\right)}$$

$$\geq \frac{\sum_{j:r_j^{(t+1)}>r_j^{(t)}} \left(A\Delta^{(t)}\right)_j^2 p_j^{(t)} \left(1 - \frac{p_j^{(t)}}{p_j^{(t+1)}}\right)}{\sum_{j:r_j^{(t+1)}>r_j^{(t)}} \left(r_j^{(t+1)} - r_j^{(t)}\right)}$$

$$\geq 26M.$$

$\qquad\square$

**Lemma C.11.** *If* ResidualSolver *returns a dual solution $r^{(T)}$ then $\mathcal{E}\left(s + \frac{MC^2}{2\left\|r^{(T)}\right\|_1} r^{(T)}\right) \geq 13M$.*

*Proof.* By Lemma C.10, we have that

$$\sum_{i=0}^{T-1} \mathcal{E}\left(p^{(t+1)}\right) - \mathcal{E}\left(p^{(t)}\right) \geq 26M \sum_{i=0}^{T-1} \left\|r^{(t+1)}\right\|_1 - \left\|r^{(t)}\right\|_1$$

leading to
$$\mathcal{E}(p^{(T)}) \geq 26M \left( \left\| r^{(T)} \right\|_1 - \left\| r^{(0)} \right\|_1 \right)$$

Since $\left\| r^{(T)} \right\|_1 \geq 2 \left( \|w\|_1 + d \right)$

$$\mathcal{E} \left( s + \frac{MC^2}{2 \left\| r^{(T)} \right\|_1} r^{(T)} \right) \geq \mathcal{E} \left( \frac{2 \left( \|w\|_1 + d \right) s}{\left\| r^{(T)} \right\|_1} + \frac{MC^2}{2 \left\| r^{(T)} \right\|_1} r^{(T)} \right)$$

$$\geq 26M \left( 1 - \frac{\left\| r^{(0)} \right\|_1}{\left\| r^{(T)} \right\|_1} \right)$$

$$\geq 13M,$$

as needed. $\qquad\square$

**Lemma C.12.** *If* ResidualSolver *returns a primal solution* $\Delta$ *then* $\|A\Delta\|_\infty \leq \frac{11}{C}$ *and*

$$\left\langle s, (A\Delta)^2 \right\rangle \leq \min_{g^\top \Delta = -1} \left\langle s, (A\Delta)^2 \right\rangle + \frac{MC^2}{2} \|A\Delta\|_\infty^2 .$$

*Proof.* If the algorithm returns $\Delta$, we have $\|A\Delta\|_\infty \leq \frac{11}{C}$, either by a solution in a single iteration, or the average over iterations when $\left\| A\Delta^{(t)} \right\|_\infty \leq S$. In both case, by Lemma C.9, and convexity, we have

$$\left\langle s, (A\Delta)^2 \right\rangle \leq \min_{g^\top \Delta = -1} \left\langle s, (A\Delta)^2 \right\rangle + \frac{MC^2}{2} \|A\Delta\|_\infty^2 .$$

$\qquad\square$

## C.2  Proof of Lemma C.6

In this section, we analyze the running time of the ResidualSolver (Algorithm 4). The running time is dominated by the linear system solves, and thus we upper bound the number of calls to the linear system solver.

We proceed similarly to the analysis of our $\ell_\infty$ regression algorithm. Let $T_{hi}$ and $T_{lo}$ be the iterations when $\left\| A\Delta^{(t)} \right\|_\infty > S$ and $\left\| A\Delta^{(t)} \right\|_\infty \leq S$, respectively before the algorithm returns a primal solution or a dual one. We also abuse the notations and denote by $T_{hi}$ and $T_{lo}$ the numbers of such iterations, respectively.

**Lemma C.13.** $T_{hi} \leq O \left( d^{1/3} \right).$

*Proof.* Again, for $t$ such that $\left\| A\Delta^{(t)} \right\|_\infty > S$, by Lemma A.2, we have

$$\frac{\mathcal{E} \left( p^{(t+1)} \right) - \mathcal{E} \left( p^{(t)} \right)}{\left\| r^{(t+1)} \right\|_1 - \left\| r^{(t)} \right\|_1} \geq 26M d^{\frac{2}{3}}.$$

Since $\left\| r^{(t+1)} \right\|_1 - \left\| r^{(t)} \right\|_1 = 1$, we get

$$\mathcal{E} \left( p^{(t+1)} \right) - \mathcal{E} \left( p^{(t)} \right) \geq 26M d^{\frac{2}{3}}$$

Therefore

$$\mathcal{E} \left( 2 \left( \|w\|_1 + d \right) s + \frac{MC^2}{2} r^{(T)} \right) \geq 26M d^{\frac{2}{3}} T_{hi}$$

Notice that $\left\| r^{(T)} \right\|_1 \leq 2 \left( \|w\|_1 + d \right) \leq 6d$ and $\mathcal{E} \left( s + \frac{MC^2}{2} \frac{r^{(T)}}{\left\| r^{(T)} \right\|_1} \right) \leq 13M \left\| r^{(T)} \right\|_1$

$$\mathcal{E} \left( 2 \left( \|w\|_1 + d \right) s + \frac{MC^2}{2} r^{(T)} \right) = 2 \left( \|w\|_1 + d \right) \mathcal{E} \left( s + \frac{MC^2}{2} \frac{r^{(T)}}{2 \left( \|w\|_1 + d \right)} \right)$$

$$\leq 6d \mathcal{E} \left( s + \frac{MC^2}{2} \frac{r^{(T)}}{\left\| r^{(T)} \right\|_1} \right)$$

$$\leq 78dM.$$

We obtain $T_{hi} \leq 3d^{1/3}$. $\qquad\square$

**Lemma C.14.** $T_{lo} \leq O\left(d^{\frac{1}{3}} \log(n)\right).$

*Proof.* Let $\alpha_j^{(t)} = \frac{r_j^{(t+1)}}{r_j^{(t)}}$. We have

$$\left\| A \frac{\sum_{t \in T_{lo}} \Delta^{(t)}}{T_{lo}} \right\|_\infty \geq \frac{11}{C}$$

We obtain that there exists a coordinate $i$ such that

$$\sum_{t \in T_{lo}} \left( A\Delta^{(t)} \right)_i C \geq 11 T_{lo}$$

We have if $\left( A\Delta^{(t)} \right)_i^2 C^2 < 100$, $\left( A\Delta^{(t)} \right)_i C < 10$, and if $\left( A\Delta^{(t)} \right)_i^2 C^2 \geq 100$, $\left( A\Delta^{(t)} \right)_i^2 C^2 = 52\alpha_i^{(t)}$. Hence,

$$\left( A\Delta^{(t)} \right)_i C \leq 10 + \mathbb{1}_{\alpha_i^{(t)} > 1} \sqrt{52\alpha_i^{(t)}}$$

We obtain

$$\sum_{t \in T_{lo}, \alpha_i^{(t)} > 1} \sqrt{\alpha_i^{(t)}} \geq \frac{T_{lo}}{\sqrt{52}}$$

For $t$ such that $\alpha_i^{(t)} > 1$, we have

$$\sqrt{\alpha_i^{(t)}} = \frac{1}{\sqrt{52}} \left( A\Delta^{(t)} \right)_i C \in \left[ \frac{10}{\sqrt{52}}, \frac{11 d^{\frac{1}{3}}}{\sqrt{52}} \right]$$

Similarly to Lemma B.4, we obtain

$$T_{lo} \leq O\left( d^{\frac{1}{3}} \log(n) \right).$$

$\qquad\square$

Lemma C.6 now follows from the above lemmas.

## D   QSC Algorithm for the Underdetermined Case

In the underdetermined case $n \leq d$, instead of using Lewis weights in the residual solver, we can simply return to the algorithm by [EV19] and use a uniform initialization of the resistances. We provide the algorithm in Algorithm 5. The analysis follows similarly (which we omit). The number of linear system solves is $O(n^{1/3} \log n)$.

## E   Approximating the Lewis Weights

Here we review the fixed point iteration by [CCLY19] for computing approximate $\ell_\infty$ Lewis weights. While the algorithm is known to provide only a one-sided approximation, this guarantee suffices for our application.

**Theorem E.1.** *On input $A \in \mathbb{R}^{n \times d}$, the algorithm* ApproxLewis$(A)$ *returns w.h.p. $\ell_\infty$ Lewis weights overestimates $w \in \mathbb{R}^n$ in the sense that*

$$w_i \geq 1_i^\top W^{1/2} A \left( A^\top W A \right)^{-1} A^\top W^{1/2} 1_i,$$
$$d \leq \|w\|_1 \leq 2d,$$

*in time $O\left( \log n \cdot \mathcal{T}_A \right)$, where $\mathcal{T}_A$ is the time to solve a linear system involving a matrix $A^\top D A$, where $D$ is a positive diagonal.*

---

**Algorithm 5** ResidualSolver$(x, M)$ for $n \le d$

---

1: **Initialize:** $r^{(0)} = 1$
2: $g_i = \frac{-1}{M}(A^\top \nabla f(x))_i$, $s_i = f''((Ax)_i)$, $t = 0$, $t' = 0$, $v^{(t')} = 0$
3: **while** $\|r^{(t)}\|_1 \le 2n$
4:      Let $p^{(t)} = 2ns + \frac{MC^2}{2}r^{(t)}$; $\Delta^{(t)} = \arg\min_{\Delta : g^\top \Delta = -1} \left\langle p^{(t)}, (A\Delta)^2 \right\rangle$
5:      **if** $\left\langle s + \frac{MC^2}{2} \frac{r^{(t)}}{\|r^{(t)}\|_1}, (A\Delta^{(t)})^2 \right\rangle \ge 13M$ **then return** $r^{(t)}$
6:      **if** $\|A\Delta^{(t)}\|_\infty \le \frac{11}{C}$ **then return** $\Delta^{(t)}$
7:      **if** $\|A\Delta^{(t)}\|_\infty \le S = \frac{11n^{\frac{1}{3}}}{C}$:
8:          Let $t' = t' + 1$; $v^{(t')} = v^{(t'-1)} + \Delta^{(t)}$
9:          **if** $\|Av^{(t')}\|_\infty / t' \le \frac{11}{C}$ **then return** $v^{(t')}/t'$
10:      $r_j^{(t+1)} = \begin{cases} \frac{1}{52} r_j^{(t)} (A\Delta^{(t)})_j^2 C^2 & \text{if } (A\Delta^{(t)})_j^2 \ge \frac{100}{C^2} \\ r_j^{(t)} & \text{otherwise} \end{cases}$
11:      $t = t + 1$
12: **return** $r^{(t)}$

---

---

**Algorithm 6** Approximate $\ell_\infty$ Lewis weights ApproxLewis$(A)$

---

1: **Input:** A symmetric polytope given by $-1_n \le Ax \le 1_n$, where $A \in \mathbb{R}^{n \times d}$
2: **Output:** Approximate $\ell_\infty$ Lewis weights $w$ such that $w_{\text{true}} \le w$ and $\sum w \le d$
3: **Initialize:** $w_i^{(1)} = \frac{d}{n}$, for $i = 1, \dots, n$. $T = 10 \log n$.
4: **for** $k = 1, \dots, T - 1$
5:      $W^{(k)} = \text{diag}\left(w^{(k)}\right)$.
6:      $B^{(k)} = \sqrt{W^{(k)}}A$.
7:      Let $S^{(k)} \in \mathbb{R}^{s \times n}$ be a random matrix where each entry is chosen i.i.d. from $N(0, 1)$, i.e. the standard normal distribution.
8:      **for** $i = 1, \dots, n$
9:          $w_i^{(k+1)} = \frac{1}{s} \left\| S^{(k)} B^{(k)} \left(B^{(k)\top} B^{(k)}\right)^{-1} \left(\sqrt{w_i^{(k)}} a_i\right) \right\|_2^2$.
10: $w_i = \frac{1}{T} \sum_{k=1}^{T} w_i^{(k)}$ for $i = 1, \dots, m$.

---

## F   Handling Affine Constraints

In this section we show that the problem we solve is in full generality, in the sense that even in the presence of affine constraints we can still minimize objectives of the type 5 without significant overheads. Formally, given an objective of the form

$$\min_{Nx=v} \sum_{i=1}^{n} f\left((Ax - b)_i\right)$$

where $A \in \mathbb{R}^{n \times d}$, and $N \in \mathbb{R}^{m \times d}$, with $m < d$, we can minimize it to high precision using the algorithms from Section 4 with minimal changes. Indeed, we can observe that, assuming the existence of an appropriate residual solver, Algorithm 3 is completely unaffected by this subspace constraint. Hence the only difficulty is posed by solving the residual problem described in Algorithm 4 while additionally enforcing the subspace constraint. Recall that this problem, in its most general form, takes as input a matrix $A$, as well as weight vectors $s, u$, and seeks an approximate minimizer for

$$\min_{\substack{x : \langle u, Ax \rangle = -1 \\ Nx = 0}} \left\langle s, (Ax)^2 \right\rangle.$$

To handle the affine constraint $Nx = 0$, we can convert this objective into an unconstrained problem via a change of variable. Indeed, let $x_0$ be a point satisfying $Nx_0 = v$. Then any $x$ in the affine space can be expressed as

$$x = x_0 + By,$$

where $B \in \mathbb{R}^{d \times \dim \ker(C)}$ and $\text{im}(B) = \ker(N)$. In other words, the columns of $B$ form a basis for the null space of $N$. Thus our problem is equivalent to

$$\min_{y:\langle u, ABy \rangle = -1} \left\langle s, (ABy)^2 \right\rangle,$$

and takes exactly the form required for the $\ell_\infty$ regression routine from Algorithm 4, which leaves us with an optimization problem over a lower dimensional space $y \in \mathbb{R}^{\dim \ker(N)}$. Unfortunately, constructing the matrix $B$ explicitly may be costly, so we want to avoid doing so. Instead, we claim that the regression algorithm can be directly executed using solvers for matrices of the type $A^\top D A$ and $N \left( A^\top D A \right)^+ N^\top$, where $D$ is a positive diagonal.

The key difficulty lies in computing the least squares step

$$x^{(t)} = By^{(t)}, \tag{11}$$

$$y^{(t)} = \arg \min_{y:\langle u, ABy \rangle = -1} \left\langle p^{(t)}, (ABy)^2 \right\rangle. \tag{12}$$

without explicitly accessing $B$.

**Lemma F.1.** *The solution for the least squares problem defined in (11) (12) can be explicitly computed as*

$$x^{(t)} = -\frac{1}{u^\top A \Delta} \cdot \Delta,$$

$$\Delta = \left( A^\top P A \right)^+ \left( -N^\top \left( N \left( A^\top P A \right)^+ N^\top \right)^+ N \left( A^\top P A \right)^+ A^\top u + A^\top u \right),$$

*where $P = \boldsymbol{diag} \left( p^{(t)} \right)$*

*Proof.* We notice that the solution to this least squares problem has solution

$$y^{(t)} = -\frac{1}{u^\top A B \left( B^\top A^\top P A B \right)^+ B^\top A^\top u} \cdot \left( B^\top A^\top P A B \right)^+ B^\top A^\top u,$$

and equivalently $y^{(t)}$ satisfies $y^{(t)} = -\frac{1}{u^\top A B z} \cdot z$, where

$$B^\top A^\top P A B z = B^\top A^\top u.$$

Therefore there exists some $w = N^\top r$ such that:

$$A^\top P A \cdot B z - A^\top u = w \in \ker \left( B^\top \right) = \text{im}(B)^\perp = \ker(N)^\perp = \text{im}(N^\top),$$

and thus we can write

$$B z = \left( A^\top P A \right)^+ \left( w + A^\top u \right),$$

wich implies that

$$N \left( A^\top P A \right)^+ \left( w + A^\top u \right) = 0,$$

and hence

$$N \left( A^\top P A \right)^+ \left( N^\top r + A^\top u \right) = 0.$$

This allows us to explicitly express

$$r = -\left( N \left( A^\top P A \right)^+ N^\top \right)^+ N \left( A^\top P A \right)^+ A^\top u,$$

$$w = -N^\top \left( N \left( A^\top P A \right)^+ N^\top \right)^+ N \left( A^\top P A \right)^+ A^\top u,$$

which finally yields

$$B z = \left( A^\top P A \right)^+ \left( -N^\top \left( N \left( A^\top P A \right)^+ N^\top \right)^+ N \left( A^\top P A \right)^+ A^\top u + A^\top u \right)$$

and

$$x^{(t)} = B y^{(t)} = y^{(t)} = -\frac{1}{u^\top A B z} \cdot z.$$

$\square$

Therefore the entire algorithm can be carried out as in the unconstrained case.

### F.1 Lewis Weights in the Affine-constrained Setting

In addition to being able to properly execute the least squared steps in Algorithm 4, one also requires a proper initialization of weights. While in general, there is no direct notion of Lewis weights in the case where affine subspace constraints are included, we can apply the same reparametrization idea. After reparametrizing the null space of $N$ in terms of the image of a matrix $B$, we obtain an unconstrained problem involving the matrix $AB$, which is the matrix for which we need to compute the $\ell_\infty$ Lewis weight overestimates we use at initialization. Again, we argue this can be done without explicit access to $B$. Note that the fixed point iteration algorithm of [CCLY19] only requires computing leverage scores of the underlying matrix $1_i^\top AB \left( B^\top A^\top PAB \right)^+ B^\top A^\top 1_i$. Since, for increased efficiency, the algorithm also uses Johnson-Lindenstrauss sketching, we further require being able to evaluate bilinear forms of the type

$$w^\top AB \left( B^\top A^\top PAB \right)^+ B^\top A^\top u \,,$$

where $P$ is a positive diagonal. As we saw before, we can evaluate

$$B \left( B^\top A^\top PAB \right)^+ B^\top A^\top u$$
$$= \left( A^\top PA \right)^+ \left( -N^\top \left( N \left( A^\top PA \right)^+ N^\top \right)^+ N \left( A^\top PA \right)^+ A^\top u + A^\top u \right) \,,$$

which yields the expression for the required bilinear form, so it can be directly applied inside the Lewis weights estimation algorithm.

