# OpenReview forum: "Quasi-Self-Concordant Optimization with $\ell_{\infty}$ Lewis Weights"
_NeurIPS.cc/2025/Conference — NeurIPS 2025 poster_

### Official Review · Reviewer_eTKg · 2025-06-30

**Clarity:** 2
**Significance:** 2
**Originality:** 3
**Rating:** 4
**Confidence:** 3

**Summary:**

This paper considers the quasi-self-concordant (QSC) minimization problem, which includes logistic regression and regularized $\ell_p$ regression as special cases. Based on the observation that QSC functions are Hessian stable, the authors propose an iterative algorithm that solves a sequence of $\ell_{\infty}$-norm constrained quadratic subproblems, reminiscent of trust-region methods. Then the authors connect their subproblem to an overdetermined $\ell_{\infty}$-regression problem. As a contribution of independent interest, they design a new $\ell_{\infty}$ regression algorithm based on iteratively reweighted least squares (IRLS) insipred by (Ene and Vladu'19). By incorporating $\ell_{\infty}$ Lewis weight overestimates, they improving the complexity from $\tilde{O}(n^{1/3}/\epsilon^{2/3})$ in (Ene and Vladu'19) to the-state-of-the-art $\tilde{O}(d^{1/3}/\epsilon^{2/3})$. This improvement directly translates to a faster algorithm for QSC minimization, particularly when $d \ll n$.

**Questions:**

## Questions
- The $\ell_2$-regularized $\ell_p$ regression problem is a key application of QSC optimization, and the authors include it in their experiments. How does the complexity result in this paper compare to existing results specifically for this setting?
- The main algorithm in Algorithm 3 involves two binary search steps over $\nu$ and $M$. While this only incurs extra logarithmic factor in the final complexity bound, it adds complexity to the implementation. Do the authors think it is possible to remove these binary search steps?

## Suggestions
- It would help readers if the authors elaborated on why their initialization using $\ell_{\infty}$ Lewis weights offers an advantage over uniform initialization. As I understand it, the main insight comes from Lemma A.2 due to (Jambulapati, Liu, Sidford'22), which provides a potentially tighter lower bound on the increase in energy.
- On a first reading, it was unclear how Section 3 relates to the central QSC optimization problem. I suggest switching the order of Sections 3 and 4 to better motivate the introduction of the $\ell_{\infty}$ regression problem.

**Ethical Concerns:**

["NO or VERY MINOR ethics concerns only"]

**Final Justification:**

During the rebuttal, the authors clarified that their convergence bounds also improve upon previous results in the context of logistic regression and regularized $\ell_{\infty}$ regression problems, particularly in the regime $d \ll n$. Given their commitment to improving the paper’s presentation, I have decided to maintain my initial rating.

**Limitations:**

yes

**Quality:**

3

**Strengths And Weaknesses:**

## Strengths
- The paper presents a simpler $\ell_{\infty}$ regression algorithm for the overdetermined setting ($d \ll n$). The proposed algorithm matches the state-of-the-art result of (Jambulapati, Liu, Sidford'22), and it is conceptually more straightforward and potentially more practical. It builds on the IRLS method from (Ene, Vladu'19), with an effective initialization using $\ell_{\infty}$ Lewis weight overestimates.
- The authors show how $\ell_{\infty}$ regression can be used as a subroutine in a trust-region framework for QSC optimization, simplifying the approach of (Adil, Bullins, Sachdeva'21) and achieving better complexity in the overdetermined regime.

## Weaknesses
- While QSC optimization covers important problems such as logistic regression and regularized $\ell_p$ regression, the paper does not clarify whether the proposed method yields concrete improvements in these specific applications.
- The exposition could be improved. In particular, compared to prior work, the two key contributions are: (1) showing that the residual subproblems in (5) can be treated as $\ell_{\infty}$ regression problems, enabling simpler IRLS-based methods; and (2) using $\ell_{\infty}$ Lewis weight overestimates to initialize the resistances to replace the dependence on $n$ with $d$. However, these points are relegated to the appendix, with limited discussion or intuition in the main text.

---

> ### Author Rebuttal · Authors · 2025-07-31
>
> We thank the reviewer their work and encouraging feedback, and the suggestions for improving the paper.  We would also like to refer the reviewer to our response to Reviewer Gv4i for the summary of our contribution.
>
> **Weakness/Questions**
>
> > The $\ell_2$-regularized $\ell_p$ regression problem is a key application of QSC optimization, and the authors include it in their experiments. How does the complexity result in this paper compare to existing results specifically for this setting?
>
> The improvement in our paper implies a direct improvement in logistic regression and regularized $\ell_{p}$ regression. More specifically, for logistic regression, the number of calls to a system solver by our algorithm is $\tilde{O}\left(d^{1/3}R\right)$ compared with $\tilde{O}\left(n^{1/3}R\right)$ by ABS21. For $\ell_{2}$-regularized $\ell_{p}$ regression, the number of calls to a system solver by our algorithm is $\tilde{O}\left(d^{1/3}Rp^{2}\mu^{-\frac{1}{p-2}}\right)$ compared with $\tilde{O}\left(n^{1/3}Rp^{2}\mu^{-\frac{1}{p-2}}\right)$ by ABS21.
>
> > Removing the binary search.
>
> Using binary search has been a commonly used approach in solving this problem. We agree with the reviewer that removing the binary search is an important direction. In practice, we can use a variant of the algorithm that updates the values of $\nu$ and $M$ during the algorithm without using binary search, and we provided such a variant in the code. However, currently we do not have a theoretical analysis for this variant and we leave it for future work.
>
> **Suggestions**
>
> We thank the reviewer for the suggestions. We will incorporate them in the next revision of the paper. We will also add further discussion on our contribution and comparison with prior work as discussed in our response to Reviewer Gv4i.

---

> > ### Comment · Reviewer_eTKg · 2025-08-05
> >
> > I thank the authors for their detailed responses to my questions and have no further questions at this time. After reviewing the discussions with Reviewer Gv4i, I concur that the paper would benefit from a clearer discussion of related work and the key novel components, to better help readers situate the contributions within the context of the existing literature.

---

> > > ### Author Response · Authors · 2025-08-09
> > >
> > > We thank the reviewer for the response. We will address the reviewers' concerns and incorporate the changes suggested by all reviewers in the revision of the paper.

---

### Official Review · Reviewer_h9Py · 2025-07-02

**Clarity:** 3
**Significance:** 3
**Originality:** 2
**Rating:** 4
**Confidence:** 4

**Summary:**

This paper considers quasi-self concordant optimization, i.e., $\min \sum_{i= 1}^n f[Ax-b]i$ , $x\in R^d, Nx = v$ where $f$ is quasi self concordant. Prior state of the art includes work by [Adil-Bullins-Sachdeva, NeurIPS’21] which solves the problem in $n^{1/3}$ iterations, each iteration solving an $n \times n$ linear system. This paper uses $\ell_{\infty}$ lewis weights and gives an algorithm that finds a solution in $d^{1/3}$ linear systems, where each system solves a $d \times d$ system.

This is a big improvement for overdetermined systems when $n>>d$. Prior such improvements have been seen for $\ell_p$ regression as in the work of JLS’22, where they used $\ell_p$ lewis weights to obtain such improvements. Furthermore, the paper gives an IRLS based algorithm inspired from EV’19 which can be implemented easily.

The algorithm is similar to the algorithm of EV’19 where the weights are replaced by overestimates of the $\ell_{\infty}$ lewis weights.

**Questions:**

Questions:
1. What is new in the analysis that allows for the incorporation of Lewis weights into EV’19? It seems like reducing to solve to $d\times d$ systems is new but I am not sure.
2. I am not sure how much easier this is than ABS’21 for simple cases of softmax regression or $\ell_{\infty}$ norm regression. Do the authors have a sense of why their algorithm might be better in practice?
3. I believe prior works on QSC optimization including ABS’21 and CJJ+’20 also work for hessian stable functions. They just seem to not explicitly write it but the proofs are done for hessian stable functions.

**Ethical Concerns:**

["NO or VERY MINOR ethics concerns only"]

**Final Justification:**

I was mainly concerned that the results were easily derivable from previous results, but based on the rebuttal it seems like it is not completely straightforward, which is why I am changing my score.

**Limitations:**

Yes

**Quality:**

3

**Strengths And Weaknesses:**

Strengths:

1. The paper gives a significant improvement over prior works in the over-constrained regime, for important regression problems.
2. The algorithm is fast in practice.

Weaknesses:

1. How much of the analysis is new is unclear as similar ideas have been used before in JLS’22 where they use a similar change in weights to recover a similar improvement to AKPS’19.

---

> ### Author Rebuttal · Authors · 2025-07-31
>
> We thank the reviewer for their work and helpful comments. We would like to refer the reviewer to our response to Reviewer  Gv4i for the clarification of our contribution in comparison with prior works. We would like to address the reviewer's concerns below.
>
> **Weakness**
>
> > How much of the analysis is new is unclear as similar ideas have been used before in JLS22 where they use a similar change in weights to recover a similar improvement to AKPS19.
>
> We would like to first clarify that the improvement that the reviewer mentioned is for $\ell_{p}$ regression problems with finite $p$ and does not apply to the QSC or the $\ell_{\infty}$ regression problems we consider in our work. JLS22 provides two very different algorithms: the algorithm the reviewer mentioned is for finite $p$ that builds on the regression algorithm of AKPS19 and does not work for $p=\infty$, and a separate algorithm for $p=\infty$  that builds on the algorithm of CJJ+20. Since it cannot handle $p=\infty$, the former algorithm cannot be combined with existing trust region templates for QSC optimization such as those used in our work or the prior work of ABS21 (see also our response to Reviewer Gv4i). Since the latter algorithm uses a completely different approach from AKPS19, it is not clear how one could leverage it for QSC optimization. We refer the reader to our response to Reviewer Gv4i for a more detailed discussion.
>
> > Q1. What is new in the analysis that allows for the incorporation of Lewis weights into EV’19? It seems like reducing to solve to $d\times d$ systems is new but I am not sure.
>
> There are two new ideas that allow for the incorporation of Lewis weights into EV19, which does not come from sparsifying the problem using Lewis weights or a direct application of the ideas from previous works. First, our algorithm differ from EV19 in the initialization using Lewis weights instead of a uniform initialization. Second, in Case 1 in the algorithm 1, we need a different update where we only update one coordinate but with a large additive step. Uniform initialization is critical in EV19's algorithm which relies on the fact that each $r_{i}$ is sufficiently large. Initialization using Lewis weights breaks this condition and thus requires a different update scheme and a new analysis.
>
> > Q2. I am not sure how much easier this is than ABS’21 for simple cases of softmax regression or $\ell_{\infty}$ norm regression. Do the authors have a sense of why their algorithm might be better in practice?
>
> Our algorithm has an iteration complexity $\tilde{O}(d^{1/3})$ while it is $\tilde{O}(n^{1/3})$ for ABS21, which we expect it will directly translate to faster running time in practice on overdetermined instances. This improvement does not come readily from prior work. For example, the improvement for softmax and $\ell_{\infty}$ regression is shown in JLS22, which is built on top of CJJ+20, which needs a complex acceleration scheme. Our work, on the other hand, provides a very simple method that does not use any acceleration, which has several advantages in practice as we discuss next.
>
> In practice, our method is robust and exhibits numerical stability. In contrast, accelerated methods require careful adjustments to their step sizes to handle precision errors, or errors from the least-squares solver (see e.g., http://blog.mrtz.org/2014/08/18/robustness-versus-acceleration.html, Devolder-Glineur-Nesterov '23), and even algorithms that have a good theoretical behavior in fixed-point arithmetic may encounter difficulties when executed on real-life floating point numerical architecture. Additionally, the algorithm by ABS21 includes several parameters ($\tau$, $\zeta$, $\alpha$). In practice, these parameters require tuning which degrade the practicality of the algorithm.
>
> > Q3: I believe prior works on QSC optimization including ABS’21 and CJJ+’20 also work for hessian stable functions. They just seem to not explicitly write it but the proofs are done for hessian stable functions.
>
> Yes, these works extend to Hessian stable objectives, as they use the same trust region template that only requires Hessian stability. We note that the main challenge is to solve the resulting subproblems. CJJ+20 provide such an algorithm for only smooth functions, and both algorithms have iteration complexity scales with $n$. Our algorithm is for general (non-smooth) functions and its iteration complexity scales with $d$ instead of $n$.

---

> > ### Comment · Reviewer_h9Py · 2025-08-04
> >
> > Dear Authors,
> >
> > Thank you for the rebuttal. I have gone through your response for Reviewer Gv4i as well and I will update my score accordingly.

---

### Official Review · Reviewer_Gv4i · 2025-07-02

**Clarity:** 3
**Significance:** 3
**Originality:** 2
**Rating:** 5
**Confidence:** 4

**Summary:**

This paper considers the problem of optimizing generalizing linear models, i.e., $\min_{x} \sum_{i} f_i(a_i^\top x – b_i)$ in the case where all the $f_i$ are the same and quasi-self-concordant (though the techniques of the paper may apply more broadly). The paper provides a trust-region based method to reduce the problem to approximately solving variants of $\ell_\infty$-regression which they solve in turn with Lewis weight overestimates used with an iteratively reweighted least squares (ILRS) algorithm. Ultimately, this provides an algorithm which solves the problem in $\tilde{O}(d^{1/3})$ iterations where $\tilde{O} (\cdot)$ here hides polylogarithmic factors as well as polynomial dependencies on a quasi-self-concordance parameter and a $\ell_\infty$ measure of the size of the effective domain. Additionally, the paper performs experiments comparing their proposed method against certain benchmarks.

**Questions:**

Question / Suggestion: The paper asserts that it is simpler than ABS21 and that JLS22 is complex. In what way is that true? I think more justification of this earlier would be beneficial. Additionally, JlS22 provided two sets of algorithms, some based on accelerated optimization and some for (p \neq infinity) that looked more like ILRS. I think the paper would benefit by more clearly comparing its algorithm to all of this prior work, e.g., making clear how challenging it is to adapt previous ILRS methods to work with the norms induced by Lewis weight over estimates in the appropriate way and how hard it is to adapt the ILRs style method in JLS22 to $\ell_\infty$

Suggestion: It would be good to clarify what QSC means on the first page and how it is used in this paper. A variety of prior work define QSC with respect to a norm (in the non-1d) case and I think it is good to clarify when the paper uses the QSC term it is very clear whether they are discussing the 1d f or the entire objective.

Suggestion: Since the experiments are done for a regularized variant of p-norm regression it might be beneficial to compare against methods tailored for that problem as well or change the problem (especially since some use similar techniques).

Here are additional detailed comments that might be helpful:

* Page 2: Line 57: “Since the linear system solve …” – there is extensive research on decreasing the cost of iterations of related methods below the time to solve a single linear system. I think this might be important to mention here.
* Page 2: Line 64: I think it might be helpful to somewhere clearly explain why the eps^{-2} in the abstract doesn’t appear in the theorem.
* Page 3: Line 130 – 131: (CR)^2/3 type rates for quasi-self-concordant functions with respect to quadratic norms were established earlier in [CJJ+20]. It seems odd to single out Doi23 as a reference for this assertion.
* Page 3: “can handle more general functions” – Is that a new feature of the method in this paper? It might be nice to clarify.

**Ethical Concerns:**

["NO or VERY MINOR ethics concerns only"]

**Final Justification:**

The discussion regarding the paper addressed some of my concerns and elevated my view of the submission. That the paper ultimate provides their results for QSC optimization I think gives a strong reason for acceptance. However, I think the depth of these results, the challenges in obtaining them, and the relationship of the approach to related work could still use careful consideration in the final version.

**Limitations:**

Yes, with the possible exception of the concerns raised elsewhere in the review.

**Paper Formatting Concerns:**

None.

**Quality:**

3

**Strengths And Weaknesses:**

Strengths: As the paper well-articulates, the problem considered encompasses a number of natural optimization problems. Consequently, it is a natural and important problem to seek improved optimization methods for. The paper provides a natural straightforward strategy, which they then implement and perform experiments on. I think this paper nicely highlights the importance of a problem which they provides an elegant solution for it while demonstrating its promise empirically. This paper seems to be making progress by promoting a natural optimization problem, providing a key theoretical benchmark for it, studying the problem in practice, seeking simpler / more straightforward algorithms, and building out the theory of ILRS. I think this paper could facilitate further research in theory and in practice on solving algorithms in this space.

Weaknesses: It is unclear how challenging it is to obtain the $\tilde{O}(d^{1/3})$ rate provided by the paper by using prior tools and the relationship of the approach to prior work could perhaps use further explanation. The paper provides a reduction from the problem to trust region problems related to $\ell_\infty$-regression and then provides a solver for both this problem and the more speciialized $\ell_\infty$-regression problem. The reduction to trust region problems seems fairly standard in light of prior work related to quasi-self-concordance that the paper cites. Additionally, as the authors note, $\ell_\infty$ regression is then solve-able more efficiently than the method the paper proposes for it (at least in terms of not having an extra $1/\epsilon^2$ factor, though potentially in a more complicated fashion). Consequently, the novelty in the paper seems more in solving the trust-region problems (as opposed to $\ell_\infty$ regression) and in providing an arguably simpler approach than what the literature gets for solving $\ell_\infty$ regression (at the cost of the $1/\epsilon^2$ factor). However, these distinctions and comparison could be better clarified and it is unclear how challenging it is to adapt prior work to solve the trust region problem. Additionally, the empirical results only compare against a limited set of methods.

Ultimately, the paper provides a new runtime for a natural quasi-self-concordant optimization problem and provides a new algorithm for $\ell_\infty$-regression. Consequently, this paper could make a great addition for NeurIPS. However, I think the writing could be improved to more clearly articulate its contributions and its relationship to prior work.

---

> ### Author Rebuttal · Authors · 2025-07-31
>
> We thank the reviewer for their work and helpful feedback. We would like to address the reviewer's concerns below.
>
> **Weakness**
> > The reduction to $\ell_{\infty}$ regression seems fairly standard in light of prior work related to quasi-self-concordance that the paper cites (and the paper “Matrix Scaling...” which is not cited but seems relevant to this reduction and may be good to discuss). As the authors note, $\ell_{\infty}$ regression is then solve-able more efficiently than the method the paper proposes for it (at least in terms of not having an extra factor, though potentially in a more complicated fashion
>
> We cite and briefly discuss this work (CMTV17) on lines 80-89. We would like to clarify that this and other prior work do not provide a reduction to $\ell_{\infty}$ regression, and there is currently no known way to directly apply $\ell_{\infty}$ regression algorithms to QSC optimization. Building on these works, we reduce the QSC problem to a sequence of subproblems where we are optimizing a quadratic function subject to an $ \ell_{\infty}$ box constraint (in addition to the linear system constraint). This problem is related to but distinct from $\ell_{\infty}$ regression and the prior work on regression is not sufficient to solve these problems even if we are only aiming for $n^{1/3}$ iterations, as we can see from e.g. ABS21. CMTV17 (and AZLOW17) provide an algorithm for these subproblems only for the specific function arising in matrix balancing, and it does not apply to general QSC functions (in addition to its iterations scaling with n instead of d). The main contribution of our work is to design a new solver for these subproblems for general QSC functions that significantly departs from ABS21 and other prior work. We would like to highlight some distinctions
>
> - Comparison with JLS22: While JLS22 introduces a new algorithm with Lewis weight for $\ell_{p}$-regression, this algorithm does not include $p=\infty$. There is a fundamental reason for this. This algorithm follows the line of works for $\ell_{p}$-regression that use the Taylor expansion to approximate the $\ell_{p}$-norm. This does not work for $\ell_{\infty}$ norm. As a consequence, for $\ell_{\infty}$-regression, JLS22 use softmax as a proxy, and Monteiro-Svaiter acceleration with a ball oracle to obtain $\tilde{O}(d^{1/3})$ rate. Monteiro-Svaiter acceleration is a complex acceleration method, and there have been multiple attempts (eg. Carmon et al. 2022) to simplify it (for example, this method requires to solve an implicit equation to find the coefficients), but the practicality of this method remains unclear. Our work on the other hand uses the dual formulation of the problem, which is fundamentally a different approach. We show a simple algorithm that incorporates Lewis weights, without any complicated acceleration scheme and allows for an efficient implementation.
>
> - Comparison with ABS21. Our work reduces the iteration complexity for solving the subproblems from $\tilde{O}(n^{1/3})$ to $\tilde{O}(d^{1/3})$ with a new insight from solving $\ell_{\infty}$ regression. As pointed out by JLS22, the approach to gain the improvement does not simply stem from sparsification with Lewis weights. Instead, this requires a new and clever way of using Lewis weights in the algorithm, which is not at all an apparent application of prior approaches. In particular, our algorithm for solving $\ell_{\infty}$-regression with Lewis weights departs from prior approaches as explained above. From the practical side, the algorithm of ABS21 uses several different parameters ($\tau,\zeta , \alpha$). In practice, this algorithm requires to tune these parameters to obtain a fast runtime, which makes it not a practical algorithm.
>
> - Comparison with EV19. While the framework of EV19 provides the basis for our algorithm, their framework requires very specifically that the solution is initialized (close to) uniformly. A significantly different initialization using Lewis weights immediately breaks EV19's convergence analysis. While being simple in hindsight, our algorithm requires a subtle use of the new energy lemma by JLS22 in a fundamentally different approach. Moreover, we cannot use the algorithm of EV19 to solve the subproblems of QSC since it is specifically designed for only pure $\ell_{\infty}$ objectives, whereas we need an algorithm that solves $\ell_{\infty} + \ell_2$ objectives
>
> > they only compare against a limited set of methods
>
> The set of implementable algorithms beyond first order methods (which are not apt for high-precision) is very limited. For reasons mentioned above and below, existing algorithms such as ABS21 do not provide a practical implementation.
>
> **Questions**
> > Q1. The paper asserts that it is simpler than ABS21 and that JLS22 is complex. In what way is that true?
>
> We would like to stress again that the algorithm by JLS22 for $\ell_{p}$-regression does not include $p=\infty$. Their algorithm for $\ell_{\infty}$-regression relies on a complex acceleration scheme. On the other hand, the algorithm by ABS21 uses a number of parameters that need to be tuned in practice, which makes it difficult to be efficiently implementable.
>
> > Q2. “can handle more general functions”.
>
> Our claim here is that our approach generalizes to $\sum f\_{i}((Ax-b)\_{i})$, where each $f_{i}$ is QSC. This generalizes the setting in ABS21 where the objective is $\sum f((Ax-b)_{i})$ for a QSC function $f$.
>
> **Suggestions and comments**
>
> We thank the reviewers for the suggestions and comments. We will incorporate them in the next revision of the paper. We will also add the above clarifications and comparison with prior work.

---

> > ### Comment · Reviewer_Gv4i · 2025-08-04
> > **Response to Rebuttal**
> >
> > Thank you to the authors for their response. I appreciate the multiple clarifications. This ultimately elevates both my view of the paper and the need for the substance of this review and others to be addressed. There are a few specific points I thought might be beneficial to respond to and discuss directly:
> > * Regarding a citation to “Matrix Scaling …” My apologies for my earlier oversight. I do indeed see that you cited it (and I believe I see why it may not come up in my search and I plan to update my review accordingly).
> > * Regarding reductions to $\ell_\infty$-regression: Indeed there was imprecision in my review, and I was using the term $\ell_\infty$-regression to both the problem of approximately minimizing convex quadratics over a box (which I’ll hence-forth refer to as the trust region problems in accordance with the submission) and to the more standard $\ell_\infty$-regression problem. My apologies for any confusion and thank you for the clarification, I may revise my review accordingly. However, I would note that the abstract of the submission states: “Our implementation of the oracle relies on solving the overdetermined $\ell_\infty$-regression problem.” In light of this discussion, is that sentence something that should instead refer to trust region problems related to $\ell_\infty$-regression or say that the implementation is related to an algorithm for overdetermined $\ell_\infty$-regression problem?
> > * I would still note that key points of my review remains: reducing quasi-self-concordant optimization as defined in the paper or $\ell_\infty$ to such trust region problems does seem standard in light of the “Matrix Scaling …paper. Additionally, while I agree that the $\ell_\infty$-regression and these box-constrained optimization are different problems, it is not clear that known state-of-the-art methods designed for one necessarily face substantial obstacles when applied to the other
> > * I find the statement “We would like to stress again that the algorithm by JLS22 for $\ell_p$-regression does not include $\ell_\infty$.” and its earlier reference a little confusing.  To confirm and make sure there is no ambiguity, JLS22 does obtain results for $\ell_\infty$ regression and the associated algorithm algorithm is closely related to one of its algorithms for $\ell_p$ regression; it is just seeming more distant to a different algorithm in JLS22 and the algorithms of the submission. A clarification if a different point was intended would be appreciated.
> > * In the discussion about the complexity of Monteir-Svaiter, the paper notes that there have been simplifications in Carmon et al. 2022 (which if I understand correctly eliminate the need to solve certain implicit equations). In light of this result, are such methods still more complex or impractical than the proposed method?

---

> > ### Author Response · Authors · 2025-08-04
> >
> > Thank you for your response.
> >
> > > Regarding the statement “Our implementation of the oracle relies on solving the overdetermined $\ell_{\infty}$-regression problem”.
> >
> > In our paper, we further reduce the subproblem (Problem 5, quadratic objective with box constraints) to an $\ell_{\infty}+\ell_{2}$ problem (Problem 6) and implement an algorithm for solving Problem 6. The implementation of the algorithm for Problem 6 is related to our algorithm for overdetermined $\ell_{\infty}$-regression problem shown in Section 3.
> >
> > > while I agree that the $\ell_{\infty}$-regression and these box-constrained optimization are different problems, it is not clear that known state-of-the-art methods designed for one necessarily face substantial obstacles when applied to the other
> >
> > While the main goal is to solve the subproblem (Problem 5, quadratic objective with box constraints), the main obstacle is to obtain a solver with $\tilde{O}(d^{1/3})$ iteration complexity instead of $\tilde{O}(n^{1/3})$. Even when we consider Problem 6 ($\ell_{\infty}+\ell_{2}$ problem), we don't think any of the current approaches is readily applicable to solving this problem. In particular, the $\ell_{\infty}$-regression solver by EV only gives $\tilde{O}(n^{1/3})$ rate and it was not known prior to our work whether we can build on it to solve Problem 6 (indeed, ABS21 based their approach on a different and more involved regression algorithm, and that extension too is far from immediate). We discuss the $\ell_{\infty}$-regression solver by JLS22 next.
> >
> > > Clarification about JLS22's algorithms
> >
> > We are not sure we understand the reviewer's question here. To clarify, JLS22 gives 2 distinct algorithms: one for $\ell_{p}$-regression ($p<\infty$) and one for $\ell_{\infty}$-regression. We cannot employ their $\ell_{p}$-regression algorithm to solve $\ell_{\infty}$-regression (the algorithm has a $O(p)^{p}$ dependency on $p$). As we mentioned in the rebuttal, their $\ell_{\infty}$-regression algorithm uses CJJ+20 which uses Monteiro-Svaiter acceleration (as a special case of QSC optimization itself). This is a completely different approach.
> >
> > > Practicality of Monteir-Svaiter from Carmon et al. 2022
> >
> > Carmon et al. 2022 provided an implementation of their scheme. The practicality is improved but the authors also mentioned specifically that it is still not competitive against Newton's method.

---

> ### Comment · Reviewer_Gv4i · 2025-08-08
> **Response to Authors**
>
> Thank you for replying. I appreciate the response and ultimately the discussions are elevating my view of the substance of the submission. However, from the responses, I am concerned about the extent to which the substance of the reviews and the source of the concerns raised will be reflected in the final version. To respond to the particular points made:
>
> * I think stating that the solver “relies on” $\ell_\infty$-regression is different than saying that it is “related to” their algorithm for $\ell_\infty$-regression and I think this is important to clarify clearly.
> * I think there is a difference between asserting that the result is not readily applicable and providing compelling evidence, e.g., a technical fact that the proofs use that seem difficult to circumvent. Again, I agreed that the prior work does not apply immediately, however it is not clear to me that the techniques underlying the prior work cannot be applied in a straightforward way. To be clear, ultimately, I think that the paper is providing an interesting approach that is great to share with the community, however I would be hesitant to assert too strongly that it overcomes too large a barrier (at least without more compelling evidence).
> * To clarify, I think what was stated in the reviewer response is incorrect. I believe JLS22 provides more than 2 distinct algorithms; it provides 2 distinct algorithms for $\ell_p$-regression as well as 1 additional algorithm for $\ell_\infty$-regression. The $\ell_\infty$-regression algorithm in the paper is indeed a MS-acceleration based method, but it is closely connected to one the paper’s algorithms for $\ell_p$ regression. Consequently, I find the assertions made by the authors about JLS22 a bit confusing or perhaps, misleading.
> * That is an interesting point about the experimental results in Carmon et al. (and perhaps worth noting when explain experimental decision in the submission), however I still feel this discussion regarding the complexity of methods and their practicality warrants more careful phrasing and consideration in the submission.
>
> I also want to add that I feel that there was at least one aspect of my previous response were not responded to, e.g., my comment about the reduction from softmax to $\ell_\infty + \ell_2$ regression being straightforward from prior work.

---

> > ### Author Response · Authors · 2025-08-09
> >
> > Thank you for your reply.
> >
> > Regarding the last question about the reduction from QSC to $\ell_\infty + \ell_2$ regression, we agree that the reduction itself is not difficult, the more difficult part is how to implement the $\ell_\infty + \ell_2$ solver.
> >
> > For the other comments, we will revise the language and clarify the issues raised by the reviewer in the revision of the paper. Again, we would like to thank the reviewer for the careful reading and insightful comments.

---

### Note · Authors · 2025-08-12

We thank all reviewers for providing valuable comments to our submission. As a final remark, we would like to summarize our responses addressing the reviewers' concerns about our paper.

- Our paper gives a new algorithm for minimizing quasi-self-concordant functions. The main contribution is a new and practical algorithm for implementing the oracle used to solve each subproblem in the trust-region method, using $\tilde{O}(d^{1/3})$ calls to a linear system solver.

- We provide a new algorithm for solving $\ell_{\infty}$-regression and the related $\ell_{\infty}+\ell_{2}$-regression problem. Our algorithm uses a simple IRLS-based method, leveraging Lewis weights in the overdetermined regime. IRLS algorithms are favored in practice (Burrus, 2012). We experimentally demonstrate the practicality of our algorithm.

- Our $\ell_{\infty}$-regression solver is fundamentally different from the algorithm by Jambulapati, Liu, Sidford (2022) which uses Monteiro-Svaiter acceleration to achieve the $\tilde{O}(d^{1/3})$ rate. Our work requires new insights to overcome technical challenges to improve the $\tilde{O}(n^{1/3})$ rate from Ene-Vladu (for $\ell_{\infty}$-regression) and Adil, Bullins, Sachdeva 2021 (for quasi-self-concordant minimization).

We appreciate the reviewers' feedback and will incorporate the clarifications on prior work, the problems we studied that we discussed in our responses, and the discussion on the specific technical challenges in the revision of our paper.

---

### Decision · Program_Chairs · 2025-09-17

**Decision:**

Accept (poster)

**Comment:**

This work studies the constrained quasi-self-concordant (QSC) minimization problem. The main contribution of the paper is a new algorithm for solving each subproblem in a trust-region method in $\tilde{O}(d^{1/3})$ calls to a linear system solver, which improves upon the previous $\tilde{O}(n^{1/3})$ calls. These results are furthermore based on a new (and arguably simpler) IRLS algorithm for $\ell_\infty$ regression which nearly achieves the same rate as the previous approach of [Jambulapati-Liu-Sidford '22] (though with an additional $\frac{1}{\epsilon^2}$ term). The discussion between Reviewer Gv4i and the authors provided much needed clarity on the particular placement on the work in relation to previous literature, and this ultimately strengthens the view of the work as a whole. All of the reviewers have expressed support for acceptance, and I also believe that this work would provide a meaningful and interesting contribution to the conference, though I strongly encourage that the authors incorporate the reviewers' suggestions that arose during the discussion.